# Diagnosis of Dextrocardia with a Pictorial Rendition of Terminology and Diagnosis

**DOI:** 10.3390/children9121977

**Published:** 2022-12-16

**Authors:** P. Syamasundar Rao, Nilesh Sai Rao

**Affiliations:** 1Children’s Heart Institute, Children’s Memorial Hermann Hospital, McGovern Medical School, University of Texas-Houston, Houston, TX 77030, USA; 2SUNY Upstate Medical University, Syracuse, NY 13210, USA

**Keywords:** dextrocardia, levocardia, dextroposition, situs solitus, situs inversus, situs ambiguous, cardiac malposition, asplenia, polysplenia

## Abstract

A significant number of patients with dextrocardia and other cardiac malpositions have other congenital heart defects (CHDs). The incidence of CHDs in subjects with cardiac malpositions is significantly greater than that in normal children, and the prevalence varies with the associated visceroatrial situs. The most useful approach to diagnosis is segmental analysis. Firstly, dextroposition should be excluded. In segmental analysis, the visceroatrial situs, ventricular location, status of atrioventricular connections, the great artery relationship, and conotruncal relationship are determined with the use of electrocardiogram (ECG), chest X-ray, and echocardiographic studies, and, when necessary, other imaging studies, including angiography. Following identification of the afore-mentioned segments, the associated defects in the atrial and ventricular septae, valvar and vascular stenosis or atresia may be determined by a review of the historical information, physical examination, and analysis of chest roentgenogram, ECG, and echocardiographic studies. Along the way, a pictorial rendition of the terminology and diagnosis of cardiac malpositions is undertaken.

## 1. Introduction

The senior author (PSR) has had an interest in dextrocardia, including other cardiac malpositions and asplenia/polysplenia syndromes since his early academic career and published papers, reviews, and book chapters on this subject [1,2,3,4,5]. The purpose of this review is to offer a pictorial rendition of the terminology and diagnosis of cardiac malpositions. Initially, the terminology used was reviewed, followed by the diagnosis. Due to the limitations of space, the management of cardiac malpositions will not be detailed.

## 2. Cardiac Malpositions

Dextrocardia and other cardiac malpositions are often associated with complicated congenital heart defects (CHDs). In addition, most physicians are confused when evaluating patients with cardiac malposition. In this paper, a systematic approach is offered to put physicians at ease when they approach patients with cardiac malposition.

## 3. Terminology

Somewhat a different terminology is used when describing cardiac positions in subjects with dextrocardia and syndromes of asplenia/polysplenia, and these will be defined before discussing the approach used for the diagnosis of these cardiac abnormalities. Some cardiologists use the apicality of the heart to determine the cardiac position, but since the apicality of the heart is difficult to discern in some patients and because apicality is not needed for the approach used, the apicality of the heart is largely ignored in this paper.

### 3.1. Levocardia

The heart is normally positioned in the left chest (Figure 1) and is designated levocardia.

### 3.2. Dextrocardia

If the heart is positioned in the right chest (Figure 2 and Figure 3), it is described as dextrocardia. The term dextrocardia should be used for conditions in which the cardiac malposition is intrinsic or primary and is not due to displacement of the heart secondary to the thoracic cage or lung abnormalities [3].

### 3.3. Mesocardia

If the heart is in the middle of the chest (Figure 4), it is described as mesocardia.

### 3.4. Dextroposition

The term dextroposition is used when the heart is in the right chest, secondary to extraneous conditions. For example, the heart is pulled to the right secondary to agenesis, hypoplasia, collapse, or surgical removal of the right lung, or it is pushed towards the right by lobar emphysema, pneumothorax, pleural effusion, or diaphragmatic hernia on the left side. The cardiac apex, if it can be identified, usually points to the left. Some examples of dextroposition are shown in Figure 5, Figure 6 and Figure 7.

### 3.5. Situs Solitus

Situs solitus is a phrase used to characterize the normal location of the visceral structures in which the liver is located on the right side of the abdomen while the stomach is on the left side (Figure 1, Figure 3, Figure 4B and Figure 5).

### 3.6. Situs Inversus

Situs inversus is a term used to characterize the reversal of the viscera from left to right. In this condition, the liver is located on the left while the stomach is placed on the right (Figure 2B and Figure 8).

### 3.7. Situs Ambiguous

Situs ambiguous is a phrase used to characterize a condition in which the situs is indeterminate and cannot be classified as situs solitus or situs inversus, usually has a midline liver (Figure 9) and may either represent bilateral right-sidedness as seen in asplenia syndrome or bilateral left-sidedness as seen in polysplenia syndrome [2,3,4]. Other names used to describe situs ambiguous are Situs Symmitricus and Situs Indeterminatus.

### 3.8. Situs Inversus Totalis

This term situs inversus totalis signifies the inversion of both heart and viscera, i.e., dextrocardia and situs inversus together (Figure 2B and Figure 8)

### 3.9. Isolated Dextrocardia

Isolated dextrocardia is a phrase used to characterize subjects with dextrocardia, but the viscera are normal in position (situs solitus), i.e., dextrocardia with situs solitus (Figure 3 and Figure 10).

### 3.10. Isolated Levocardia

The term isolated levocardia denotes that the heart is on the left side (levocardia)—normal, but the viscera are reversed left to right (situs inversus) (Figure 11).

## 4. Prevalence

The prevalence of dextrocardia is estimated to be 0.12 per 1000 new patients of all age groups, while it is slightly higher at 0.29/1000 in neonates. In patients with dextroposition, the incidence of CHD is similar [6] to that seen in the normal population, which is 8 per 1000 live-born babies [7,8,9]. However, in patients with dextrocardia, the prevalence of CHDs is higher than that seen in normal populations and differs with the accompanying status of the visceroatrial situs [4,5]. In subjects with situs inversus totalis, the incidence of CHD is 3 to 10% [6], while it is almost 100% in babies with isolated dextrocardia and isolated levocardia [3]. As mentioned above, this is in contrast to the CHD prevalence of less than 1% in the otherwise normal population.

## 5. Diagnostic Method for Cardiac Malposition

This diagnostic approach was initially developed [3] by the senior author (PSR), incorporating some of the principles endorsed by van Praagh [10,11,12,13], Arcilla [14], Lev [15,16], Campbell [17,18], de la Cruz [19], Shinebourne [20], Stanger [21] and their associates and was subsequently modified in more recent reviews [4,5]. The diagnostic approach is similar irrespective of the cardiac position, namely, dextrocardia, mesocardia, or levocardia. In the majority of patients, physical examination, chest roentgenogram, electrocardiogram (ECG), echocardiogram, and, more recently, magnetic resonance imaging (MRI) or computed tomography (CT) investigations are adequate to come up with a reasonably accurate diagnosis. Cardiac catheterization with selective cineangiography is not usually necessary unless it is an integral part of a catheter-based interventional procedure. The location of the heart on the right side of the chest may be better determined by palpation of the cardiac impulse than by auscultation of the heart sounds. Of course, a chest X-ray is confirmatory of cardiac malposition. In this review, electrocardiographic, echocardiographic, and angiographic examples will be presented, and MRI and CT images are not shown. The approach used is to answer the following questions:Why is the heart in the right side of the chest?Where are the atria located?Are there two ventricles or one? If two, where are the ventricles located in relation to each other?What is the status of the atrioventricular connections?How are the great arteries related to each other and the ventricles?What is the conotruncal relationship?

### 5.1. Why Is the Heart in the Right Side of the Chest?

The first step is to determine the reason why the heart is in the right chest; is it due to intrinsic malposition of the heart (true dextrocardia) (Figure 2, Figure 3 and Figure 8, Figure 9 and Figure 10), or is it due to being pushed or dragged to the right side by abnormalities of the thoracic cage or pulmonary pathology (dextroposition) (Figure 5, Figure 6 and Figure 7, Figure 12, Figure 13, Figure 14 and Figure 15). Careful inspection of the tracheal position usually reveals that the trachea is shifted to the right in patients with dextroposition of the heart (Figure 5, Figure 6, Figure 7 and Figure 15), while the trachea is close to the midline in subjects with innate dextrocardia (Figure 2, Figure 3, Figure 8, Figure 9 and Figure 10). The apex of the heart is usually pointed towards the left in dextroposition patients. In addition, chest X-ray will show bony thoracic abnormality (Figure 12) or pulmonary pathology (Figure 5, Figure 6, Figure 7, Figure 12, Figure 13, Figure 14 and Figure 15). Examples of thoracic cage abnormality (Figure 12), empyema (Figure 13), pneumothorax (Figure 6), congenital pulmonary cyst in the left lung (Figure 14), lobar emphysema on the left or left side diaphragmatic hernia (Figure 7) pushing the heart to the right and right lung collapse (Figure 15), right pneumonectomy or hypoplastic right lung (Figure 5), as well as that caused by scimitar syndrome (Figure 5), pulling the heart to the right, are shown in Figure 5, Figure 6, Figure 7, Figure 12, Figure 13, Figure 14 and Figure 15.

The reasons for taking this step are two-fold; first, the prevalence of associated CHD is vastly different between both groups (see the section on Prevalence), and second, most commonly, the pulmonary pathology seen in dextroposition needs to be addressed on a relatively urgent basis.

### 5.2. Where Are the Atria Located?

Following the exclusion of dextroposition and confirmation that the position of the heart in the right chest is due to intrinsic cardiac malposition, the site of the atria should be determined. There is a consistent relationship between the atrial site and abdominal viscera (as exemplified by liver and stomach), described as viscero-atrial concordance [2,3,23]: a left-sided stomach and right-sided liver are seen with morphologic left atrium (LA) on the left side and morphologic right atrium (RA) on the right side, meaning situs solitus. Similarly, a left-sided liver and right-sided stomach are seen with right-sided morphologic LA and left-sided morphologic RA, meaning situs inversus. Cardiac malposition may be seen with 1. Situs solitus. Normal relationship of the viscera and atria (right to left), 2. Situs inversus. Left-to-right reversal of the viscera and atria, or 3. Situs ambiguous with indeterminate situs. The situs ambiguous is usually accompanied by heterotaxy syndromes (asplenia and polysplenia). The situs of the atria may be investigated by means of several principles/methodologies.

#### 5.2.1. P Waves in the Electrocardiogram (ECG)

In normal individuals, the cardiac electrical impulse is initiated in the sino-atrial (SA) node. The SA node is situated at the junction of the superior vena cava and RA. The cardiac impulse travels to the left and inferiorly [24]. The ensuing depolarization of the atria results in P waves of the ECG tracing. The resultant P wave is positive in both leads I and AVF with a P wave axis of approximately +45° (Figure 16 and Figure 17). Such normal P waves are indicative of situs solitus of the atria with the RA on the right side and the LA on the left side.

If the P waves, on the other hand, are negative in lead I and positive in lead AVF (Figure 18) with a P wave axis of approximately +135° (Figure 16), the atria are likely to be inverted with morphologic LA on the right and morphologic RA on the left, i.e., situs inversus.

If the P waves are negative in lead AVF and positive in lead I (Figure 19), giving an axis is −45° (Figure 16), it is generally termed low atrial or coronary sinus rhythm. Such a P wave axis is not useful in assessing the situs of the atria. Nevertheless, such coronary sinus (low atrial) rhythms are frequently seen in subjects with infrahepatic interruption of the inferior vena cava (IVC) and persistent left superior vena cava; these venous anomalies are often associated with heterotaxy syndromes (asplenia/polysplenia).

ECG is frequently obtained during a routine investigation of any CHD, and review of the P wave morphology in the ECG is an easy and inexpensive way of identifying atrial situs in patients with cardiac malposition.

#### 5.2.2. Visceroatrial Concordance

The principle of visceroatrial concordance [23], as stated above, dictates that a liver on the right side and stomach on the left side (Figure 1, Figure 3, Figure 4B and Figure 10) are suggestive of situs solitus, i.e., the morphologic LA is on the left side, and the morphologic RA is on the right side. Similarly, a right-sided stomach and a left-sided liver (Figure 2B, Figure 8 and Figure 11) are indicative of situs inversus, i.e., the morphologic RA on the left and the morphologic LA on the right.

To utilize the principle of visceroatrial concordance, the locations of the liver (by its whitish opacity) and stomach (by its black gaseous opacity) should be visualized in posteroanterior view of chest radiographs (Figure 1, Figure 2B, Figure 3, Figure 4B, Figure 8, Figure 10 and Figure 11). Occasionally, the stomach bubble may not be present on routine chest X-rays (Figure 20A). In such situations, reviewing all the available chest X-rays (Figure 20B) or even injecting small quantities of air or barium (Figure 21) via a nasogastric tube may be warranted. However, the authors have not resorted to this since some chest X-rays exhibited a gaseous stomach bubble (Figure 20B).

If the lobes of the liver are of equal size or if the liver is located in the midline (Figure 2A and Figure 9), regardless of the location of the gaseous opacity of the stomach, situs ambiguous is likely, and the possibility of heterotaxy syndromes (asplenia/polysplenia) exists [2,3]. The rule of visceroatrial concordance appears to have a greater degree of dependability than the P wave axis in an ECG in accurately detecting the situs of the atria [3,4,5]. Although this well-established principle of visceroatrial concordance is fairly helpful, it has been found to be misleading in a few cases [25].

#### 5.2.3. Tracheobronchial Tree Pattern

In typical patients with situs solitus, the bronchus on the right side is short and wide and descends somewhat steeply, whereas the bronchus on the left side is longer and narrower than the right bronchus and descends rather horizontally (Figure 22A and Figure 23). On the contrary, in patients with situs inversus, the tracheobronchial tree configuration is inverted (Figure 22B and Figure 24) [2,3,4,5]. The tracheobronchial tree pattern seems more correct than the above two approaches to identifying the atrial situs [2,26,27]. Tomography has been used in the past [27] to more accurately determine bronchial morphology and measure the bronchial lengths but is not routinely used at the present time because of increased radiation exposure associated with tomography and the availability of other imaging studies.

If both bronchi (right and left) have the appearance of morphologic right bronchi (Figure 21 and Figure 22C), asplenia syndrome is likely to be present, whereas morphologic left bronchi on both sides (Figure 22D and Figure 25) is indicative of polysplenia syndrome [2,3,26,27]. Exceptions to these observations have been seen but are uncommon [3,28,29].

#### 5.2.4. Vena Cava–Aorta Relationship

A predictable relationship between the locations of the aorta (Ao) and the IVC at the level of the diaphragm with the atrial situs seems to exist [4,5,30]. The location of the IVC and Ao at the level of the diaphragm can easily be imaged on short-axis echo views (Figure 26A). The IVC is typically larger than the Ao (Figure 26A). In addition, the Doppler flow patterns with the venous flow for the IVC (Figure 26B) and arterial flow for the Ao (Figure 26C) confirm the identity of the respective vessel.

In patients with situs solitus, the Ao is on the left of the spine, and the IVC is on the right of the spine (Figure 26). In subjects with situs inversus, the positions of the IVC and Ao are reversed, left to right, with the IVC on the left side and Ao on the right side. In subjects with dextro-isomerism (asplenia syndrome), the IVC and Ao are placed together on either the right or left side of the spine. The Ao is usually anterior to the IVC. In subjects with levo-isomerism (polysplenia syndrome), the Ao is typically in the midline, anterior to the spine, and the azygos vein is located behind the aorta. The location of the azygos vein is on the right of the spine in subjects with azygos continuation of the interrupted infrahepatic IVC, while its location is on the left side of the spine in subjects with hemiazygos continuation of the interrupted infrahepatic IVC. Consequently, the comparative locations of the Ao and IVC are valuable in the assessment of the atrial situs [4,5,30].

#### 5.2.5. Venoatrial Concordance

During embryonic development, the IVC is connected to the sinus venosus, which later (post-natal) becomes the RA. Consequently, the side of the IVC determines the location of the RA. Therefore, the IVC is on the right side in subjects with situs solitus, while the IVC is on the left side in subjects with situs inversus. The position of the IVC can easily be defined by echo imaging in the subcostal view (Figure 27). The absence of IVC entrance into the right atrium in subjects with interrupted IVC (infrahepatic) with azygos or hemiazygos can also be shown in careful subcostal echo studies.

More invasive techniques, such as nuclear angiography or cardiac catheterization (Figure 28), may also define the atrial situs but are not required for the sole purpose of atrial situs determination. In subjects who have infrahepatic interruption of the IVC with either azygos or hemiazygos continuation, the position of the azygos vein is not useful in determining the atrial situs.

#### 5.2.6. Atrial Morphology

Atrial morphology may be assessed by transesophageal echocardiography (TEE), selective atrial angiography, or by surgical inspection. If these procedures are performed for some other reason, such data may be used to determine the atrial situs. The atrial appendages have characteristic shapes in that the RA appendage is large and broad, while the LA appendage is narrow, tubular, and small. If TEE, selective atrial angiography (Figure 29), or surgical inspection is undertaken for any other reason, securing such data at that time is helpful in evaluating the atrial situs.

If the above methods of atrial situs evaluation are at variance with each other, asplenia/polysplenia syndromes (heterotaxy) should be suspected. Other features that support heterotaxy are 1. Symmetric bronchial pattern (Figure 22C,D), namely, bilateral morphologic right bronchi (dextroisomerism) (Figure 21 and Figure 22C) or bilateral morphologic left bronchi (levoisomerism) (Figure 22D and Figure 25), 2. The divergence between visceroatrial situs and position of the heart, i.e., isolated dextrocardia (Figure 3 and Figure 10) or isolated levocardia (Figure 11 and Figure 21), 3. Midline or symmetric liver on X-ray (Figure 2A and Figure 9), 4. Roentgenographic, echographic or angiographic evidence for infrahepatic interruption of the IVC, and 5. Evidence for malrotation of the intestine (Figure 10 and Figure 11). Blood smear for Howell-Jolly bodies & Heinz bodies, barium gastrointestinal series to detect malrotation of the midgut, abdominal ultrasound to identify the spleen and liver, radio-isotopic scanning of the liver and spleen and rarely, selective angiography may help clarify the existence of asplenia/polysplenia syndrome. Further discussion of these topics is outside the scope of this article, and the interested reader may review prior publications [2,3,4,5] for details.

### 5.3. Are There Two Ventricles or One? If Two, Where Are the Ventricles Located in Relation to Each Other?

In theory, patients with cardiac malposition may have one (single) or two ventricles. The distinction between one and two ventricles is feasible by echocardiography (Figure 30) and angiography (Figure 31 and Figure 32).

In case there are two ventricles, the question to address is: are the ventricles normally positioned with the right ventricle (RV) on the right side and the left ventricle (LV) on the left side? Or, are the ventricles inverted with morphologic RV on the left side and morphologic LV on the right side? Several methods have been used to make this assessment:

#### 5.3.1. Electrocardiogram

It is imperative to record right chest leads (RV5 and RV6) in addition to the usual left chest leads to make an adequate interpretation of ECG of dextrocardia patients.

##### QRS Morphology

In normal children (or even in adults), the pattern of QRS complex, namely, an rS pattern in the leads V1 and V2, is indicative that the underlying ventricle is RV and a qRs pattern in the leads V5 and V6 is suggestive that the underlying ventricle is LV [32,33,34]. Such a logic of using QRS patterns to assess the ventricular location does not apply to cardiac malpositions [3,4,5] because most patients with cardiac malposition have RV hypertrophy or a single ventricle in addition to varying degrees of rotation of ventricular chambers.

##### Initial QRS Vector

While QRS morphology is not helpful, the initial QRS vector may be more useful. In normal hearts, the initial component of the QRS complex is derived from the depolarization of the ventricular septum. Normally, although the interventricular septum is depolarized from both the left and right sides, the left portion of the interventricular septum is activated a bit sooner than the right septal component. The ensuing initial electrical activity is directed to the right, anteriorly, and slightly superiorly [24]. Such depolarization will produce q waves in the left chest leads, no q waves in the right chest leads, and a q wave in lead AVF (Figure 17).

In subjects with normally related ventricles, i.e., the RV on the right side and the LV on the left side, the initial ventricular (septal) depolarization produces q waves in leads V5 and V6 without a q wave in leads V1 and V2. (Figure 17). This is irrespective of cardiac position (dextrocardia, levocardia, or mesocardia)

In patients with reversed ventricles, i.e., the morphologic LV on the right side and the morphologic RV on the left side, the conduction system is also inverted, following the rule that the conduction system follows the ventricles. This will result in q waves in the right chest leads and no q waves in the left chest leads (Figure 19 and Figure 33).

Though the examination of the initial QRS vector has a sound theoretical basis, the usual occurrence of either RV hypertrophy or single ventricle and varying amounts of rotation of the cardiac structures suggests that such an analysis is not completely dependable in firming up of the ventricular location.

#### 5.3.2. Echocardiography and Angiography

Several characteristic anatomical features of the ventricles themselves, the relationship of the atrioventricular (AV) to semilunar valves, and attachments of AV valves to the septum are useful in defining relative ventricular location. Other established principles/rules that are helpful in ventricular localization will also be reviewed in this section. These can be evaluated by echocardiographic and angiographic examination.

##### Ventricular Trabeculations and Shape

The morphologic LV is a smooth-walled structure with fine trabeculations and a foot-shaped appearance, whereas the morphologic RV has coarse trabeculations and a triangular shape. These features are demonstrated by echocardiographic (Figure 34) and angiographic (Figure 31, Figure 35 and Figure 36) studies. It should be noted that the characteristic trabecular pattern of the ventricles is seen irrespective of great vessel relationship: normally related great vessels (Figure 31 and Figure 34), transposed great arteries in levocardia (Figure 35) or transposed great arteries in dextrocardia (Figure 36).

##### Atrioventricular Valve-to-Semilunar Valve Relationship

It has been established that AV valves go with the respective ventricular chambers in that the mitral valve is an essential part of the LV while the tricuspid valve is an integral part of the RV [10,11,13,28]. The morphologic LV has little or no conus musculature, and consequently, the mitral valve and aortic valve are in fibrous continuousness with each other (Figure 35A,B, Figure 36A and Figure 37A,B). However, in the morphologic RV, a muscular structure (crista supraventricularis) separates the tricuspid valve from the pulmonary valve, and therefore, fibrous continuity between the AV valve and semilunar valve (Figure 31A, Figure 35C,D, Figure 36B and Figure 37C) cannot be demonstrated. These features may be demonstrated in echocardiography (Figure 37) and angiography (Figure 31, Figure 35 and Figure 36).

##### Atrioventricular Valve Attachments to the Septum

The attachments of the AV valve leaflets to the interventricular septum have a characteristic pattern. In subjects with the normal left to right ventricular relationship, attachment of the tricuspid valve to the interventricular septum is at a lower level than that of the medial leaflet of the mitral valve (Figure 38A,B). On the contrary, in patients with ventricular inversion, the valvar attachments to the interventricular septum are reversed with the right-sided AV (morphologic mitral) valve attachment higher than the left-sided AV (morphologic tricuspid) valve attachment (Figure 39).

##### Loop Rule

The principle of the loop rule [3,4,5,10,11,23] dictates that the relationship of semilunar valves is predictive of the looping of the ventricles (d-loop or l-loop), and the loop status, in turn, helps to indicate the site of the ventricles. The loop rule is formulated on the basis of the embryonic development of the heart. In the 3 to 4 week-embryo (when the embryo is about 2.2 mm long), the developing cardiac structure is a straight tube in the pericardium [37,38]. The heart tube grows faster than the pericardium. Due to this length discrepancy during normal growth and development, the cephalic part of the heart tube is forced to bend; it usually bends in a ventral and caudal way and right-ward [23,37,38]. This right-ward direction is a d-loop which results in the placement of the bulbus cordis (future RV) to the right and the embryonic ventricle (future LV) to the left. On the contrary, if the cardiac tube bends to the left side, an l-loop occurs, in which case, the morphologic RV is located on the left side while the morphologic LV is placed on the right side [3,4,5,10,23,37,38].

Based on the loop rule, the aortic valve to the right side of the pulmonary valve indicates a d-loop, and d-loop predicts that the RV is located on the right side and the LV on the left side. Similarly, the aortic valve positioned to the left of the pulmonary valve suggests an l-loop, which suggests that the morphologic RV is on the left while the morphologic LV is on the right (inverted) [3,4,5,10,23,37,38]. All the features described above can be defined by both echocardiography and selective cine angiography and are helpful in evaluating the looping status. Examples of d-loop in subjects with normally related great vessels (Figure 31), those with d-transposition of the great vessels (Figure 35) and those with dextrocardia (Figure 36), and those with l-loop illustrating inverted ventricles (Figure 40 and Figure 41) are shown.

Coronary artery anatomy also has a predictive value in determining the looping status: the left anterior descending (LAD) coronary artery originates from the left coronary artery (LCA) in patients with d-loop, while the LAD originates from the right coronary artery in subjects with l-loop. Although these may be defined by carefully performed echo studies, the anatomy of the coronary artery is better defined by angiography.

Finally, chirality (right or left-handedness) has been used to assign loop status and, thus, ventricular localization [40]. When the direction of the RA to RV and RV outflow has a right-hand topology, it is likely to be a d-loop with RV on the right side and LV on the left side. If the RA to RV and RV outflow has a left-hand topology, an l-loop is expected with RV on the left side and LV on the right side.

### 5.4. What Is the Status of Atrioventricular Connections?

After the visceroatrial situs and the ventricular locations are defined, the relationship between atria and ventricles should be assessed. These relationships are: concordant, with the RA emptying into the RV and the LA emptying into the LV (Figure 30A), and discordant, with the RA emptying into the morphologic LV and the LA emptying into the morphologic RV (Figure 39). Other AV connection abnormalities are: both the right and left atria emptying into a single ventricle (double-inlet left ventricle (Figure 30B)), both atria emptying into both ventricles via one common AV valve in the form of AV septal defect (Figure 42), a common atrium emptying into a single ventricle via a single AV valve (the so-called cor biloculare) (Figure 43), a common atrium emptying into inverted ventricles via a single AV valve (Figure 44) and atresia of either tricuspid or mitral valve (Figure 45). In addition, straddling or overriding of the AV valve over the ventricular septum may also occur. Such abnormalities may be defined by methodical echocardiographic imaging with the rare need for MRI, CT, and angiographic studies.

### 5.5. How Are the Great Arteries Related to Each Other and to the Ventricles?

The relationship between the great arteries is characterized as 1. Normal, in which case, the aortic valve is placed inferior to, posterior to, and to the right side of the pulmonary valve (Figure 31 and Figure 46), 2. Transposed great arteries (d-TGA) with the aortic valve superior to, anterior to, and to the right side of the pulmonary valve (Figure 35, Figure 36 and Figure 47), or 3. Inverted (l-TGA) with the aortic valve superior to, anterior to, and to the left side of the pulmonary valve (Figure 40 and Figure 41). If the inter-relationship between the semilunar valves is abnormal but cannot be classified into one of the above groups, such abnormalities may be designated as malposition (d-malposition, l-malposition, AP-malposition).

The ventricle to great artery (Ao and PA) relationship may be classified into 1. Concordant with the RV giving origin to the PA and the LV giving origin to the Ao, as seen in normal individuals (Figure 31 and Figure 46), 2. Discordant with A. the RV giving origin to the Ao and the LV giving origin to the PA, i.e., d-TGA in levocardia (Figure 35 and Figure 47) as well as d-TGA (d loop) in dextrocardia (Figure 36) and B. the morphologically RV giving origin to the Ao and the morphologically LV giving rise to the PA (l-TGA with l-loop) (Figure 40 and Figure 41), 3. double-outlet ventricle with A. both great vessels arising from the RV, i.e., double-outlet right ventricle (DORV) (Figure 48 and Figure 49) or B. both great arteries from the LV, i.e., double-outlet left ventricle (DOLV).

Other ventriculo-arterial connection abnormalities are only one great vessel arising for the ventricles, i.e., truncus arteriosus (Figure 50), and atresia of one of the semilunar valves/great vessels, namely pulmonary and aortic atresia. As reviewed for the other anomalies in the preceding sections, these cardiac anomalies can also be defined by echocardiography with an infrequent necessity for other imaging tools, namely, MRI, CT, and angiography.

### 5.6. What Is the Conotruncal Relationship?

The position of the conal tissue relative to the pulmonary and aortic valves is characterized as subpulmonary or subaortic, or it may be present beneath both the semilunar valves, or it may be absent bilaterally. The conal tissue displaces the semilunar valve anteriorly and superiorly; the more conus tissue underneath a semilunar valve, the more superior and anterior that semilunar valve is displaced. Therefore, superior to inferior and anterior-to-posterior relationships of the semilunar valves indicate conal anatomy. In normal subjects, the conus is subpulmonary (Figure 31 and Figure 37C). In patients with transposition, subaortic conus is seen whether it is d-transposition (Figure 35 and Figure 36) or l-transposition (Figure 40, Figure 41 and Figure 51). Bilateral conus is seen in DORV (Figure 48B and Figure 49B), and the conus is absent in the double-outlet left ventricle. The conal tissue and superior and anterior dislocation of the semilunar valves can be easily ascertained by echo imaging (Figure 31, Figure 35, Figure 36, Figure 37C, Figure 40, Figure 48B, Figure 49B and Figure 51) and may also be evaluated similarly if other imaging studies are performed.

### 5.7. Summation

Whereas atrial situs can be successfully appraised with routine chest roentgenogram and ECG, the ventricular relationship, atrioventricular connections, great artery, and conotruncal relationship need echocardiography and other imaging studies for accurate evaluation. After the above issues are addressed, the existence of other abnormalities such as atrial or ventricular septal defects, pulmonary or aortic stenosis/atresia, and transposition of the great vessels are made by usual methods. Anomalies of the systemic venous and pulmonary venous connections should also be scrutinized.

## 6. Therapy

Cardiac malposition per se does not need any treatment; instead, the associated cardiac defects and the resulting hemodynamic disturbance should be addressed. Similarly, dextroposition of the heart does not need any treatment, but the associated pulmonary pathology should be treated. Due to the long length of this paper, these treatment aspects will not be addressed in this paper.

## 7. Summary and Conclusions

Cardiac malpositions are frequently associated with significant CHDs. The CHD prevalence is pointedly higher than that seen in normal children, and the heart defect prevalence differs with the associated viscero-atrial situs; babies with isolated dextrocardia and isolated levocardia have the highest incidence of CHD. In the authors’ opinion, segmental analysis is the best approach to making a correct diagnosis. However, dextroposition should be excluded first. In this method of diagnosis, the visceroatrial situs, location of the ventricles, status of atrioventricular connections, the relationship of the great vessels, and conotruncal relationship are ascertained by utilizing ECG, chest roentgenogram, and echo-Doppler studies and when needed other imaging studies, including cineangiography. Following the identification of the atrial, ventricular, and great arteries sites, associated septal defects, obstructive lesions of the valves and vascular structures are diagnosed by evaluation of the historical information, cardiovascular examination, and scrutiny of chest roentgenogram, ECG, and echocardiographic and other imaging examinations. While the cardiac malposition itself does not require treatment, addressing the pathophysiologic aberration resulting from the defect complex is necessary and was not addressed in this paper because of space limitations.

## Figures and Tables

**Figure 1 children-09-01977-f001:**
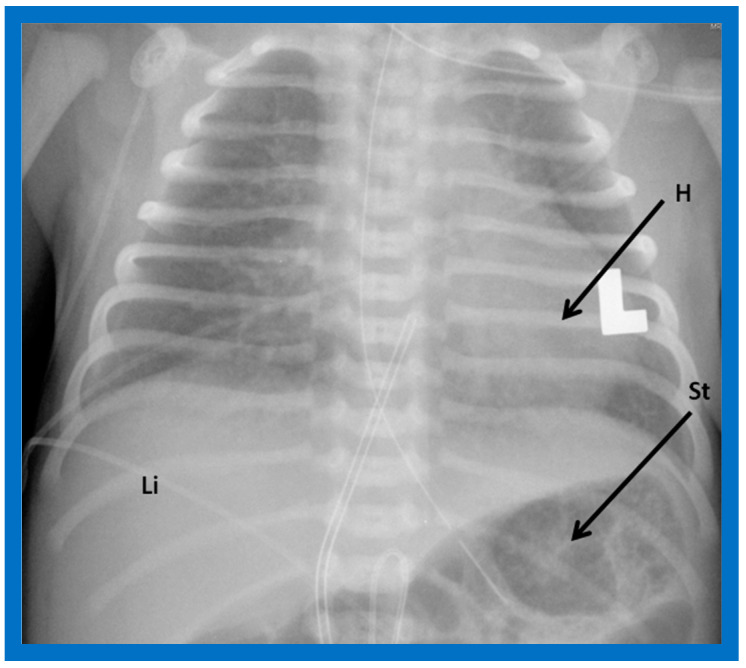
Chest roentgenogram of a baby demonstrating the heart (H) in the left chest (L), i.e., levocardia. This is the usual position of the heart in most patients. The roentgenogram further demonstrates the liver (Li) on the right side of the abdomen and the gaseous opacity of the stomach (St) on the left side, again normal. This is described as situs solitus of the viscera. L indicates left. Reproduced from reference [4].

**Figure 2 children-09-01977-f002:**
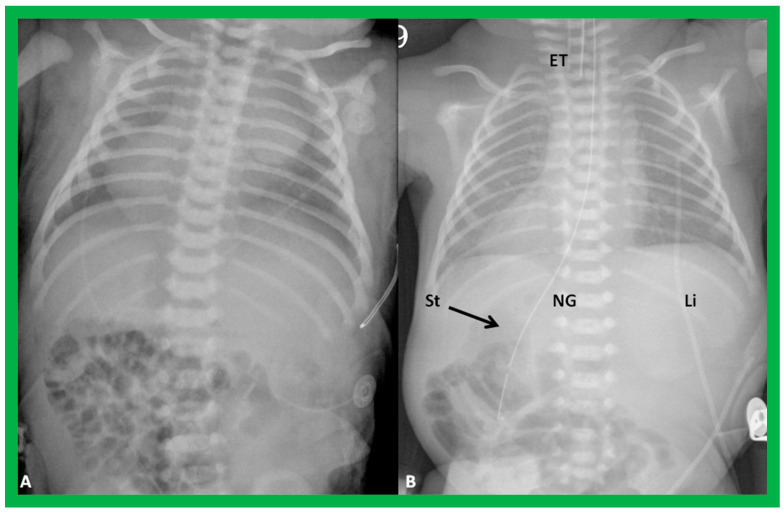
Chest X-rays of two different newborn babies with hearts situated in the right chest, i.e., dextrocardia. (**A**) The liver (Li) is visualized across the abdominal cavity without seeing the stomach air bubble; thus, it is not possible to establish the visceral situs. (**B**) On the other hand, the stomach (St) air shadow is faintly seen on the right side of the abdomen while the Li is imaged on the left side. The findings in (**B**) indicate inversion of the visceral position, situs inversus. Furthermore, the location of the St on the right is established by the location of the NG (nasogastric) tube. Air in the intestine is seen in the lower portion of the abdomen in both infants. ET, endotracheal tube. Reproduced from reference [4].

**Figure 3 children-09-01977-f003:**
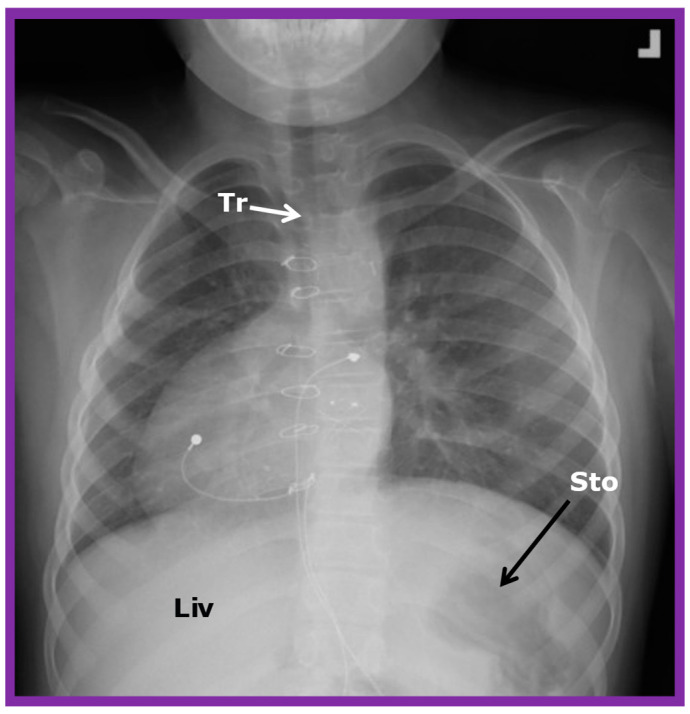
Chest X-ray of a child demonstrating the heart in the right chest (dextrocardia). The positions of the liver (Liv) on the right side, and stomach (Sto) on the left side are seen, indicative of visceral situs solitus. This is described as isolated dextrocardia. Tr. Trachea; inverted L, indicates left. Sternal and pacemaker wires (not labeled) are seen and are related to prior surgery. Modified from reference [5].

**Figure 4 children-09-01977-f004:**
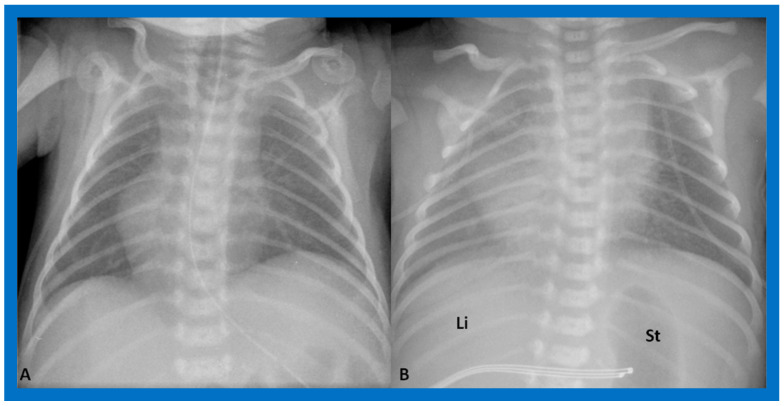
Chest X-rays of two neonates with their hearts in the middle of the chest, mesocardia. (**A**) The liver is visualized transversely in the abdomen; no gas bubble of the stomach (St) is seen. Therefore, visceroatrial situs cannot be determined with these findings. (**B**) The gaseous opacity of the St is visualized on the left side of the abdomen, and the liver (Li) is seen on the right, suggesting situs solitus. Reproduced from reference [4].

**Figure 5 children-09-01977-f005:**
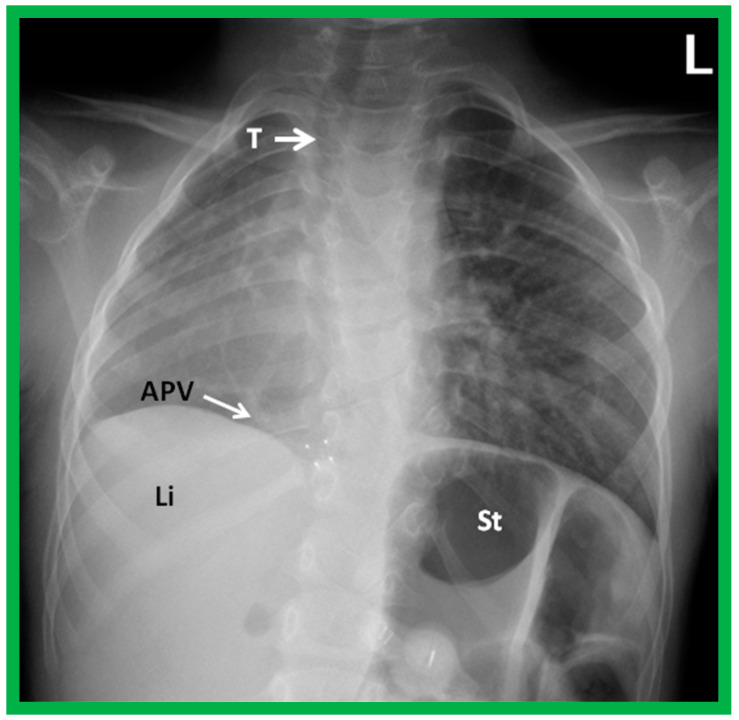
A chest roentgenogram of a child with dextroposition of the heart related to Scimitar syndrome. Hypoplastic right lung pulls the heart to the right. Note that the trachea (T) is displaced to the right, related hypoplastic right lung. An anomalous pulmonary vein (APV) is pointed out by an arrow; the name, Scimitar syndrome is derived from the sickle shape of the APV. Li, liver; St, stomach. Reproduced from reference [4].

**Figure 6 children-09-01977-f006:**
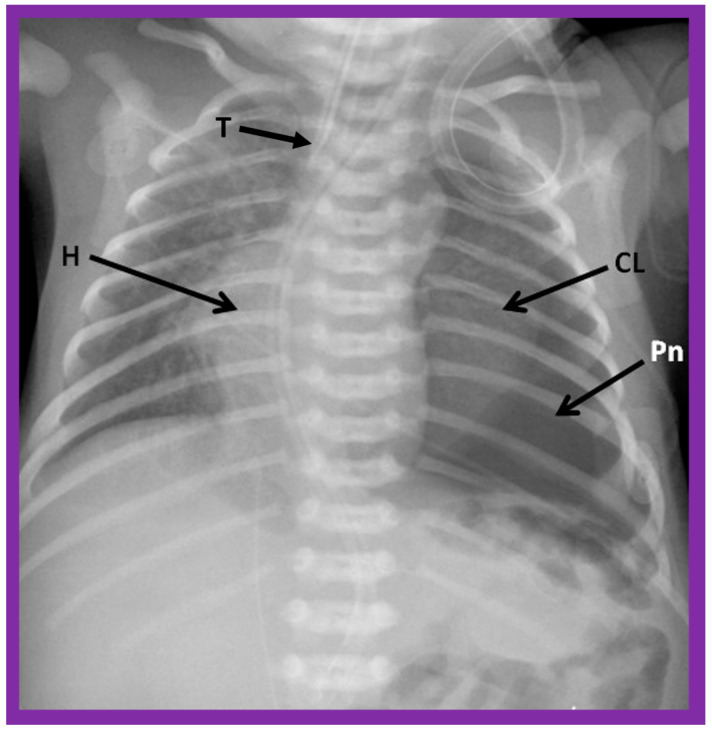
A chest roentgenogram of a baby illustrating dextroposed heart (H) due to left pneumothorax (Pn) pushing the heart into the right side of the chest. A collapsed left lung (CL) is pointed out. The trachea (T) is seen to deviate to the right. Modified from reference [4].

**Figure 7 children-09-01977-f007:**
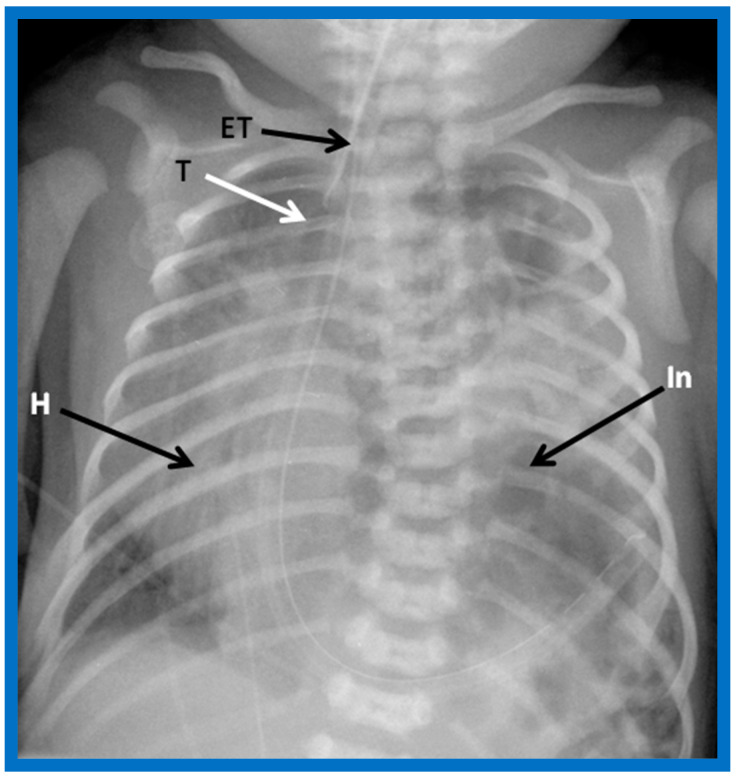
A chest roentgenogram of an infant demonstrating dextroposed heart (H) secondary to diaphragmatic hernia on the left side. The abdominal contents, including intestines (In) are relocated to the left side of the chest because of the diaphragmatic defect, pushing the heart into the right chest. The trachea (T) is displaced to the right. ET, endotracheal tube. Reproduced from reference [4].

**Figure 8 children-09-01977-f008:**
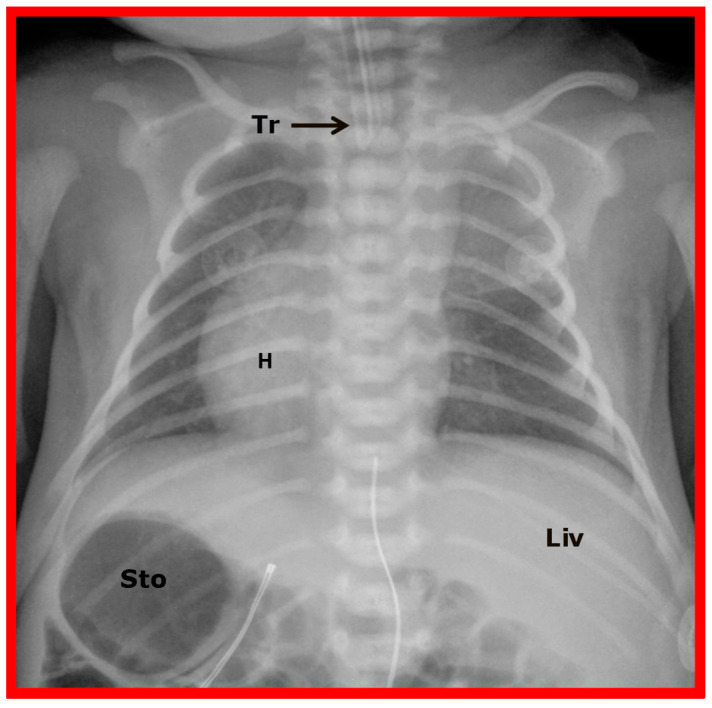
A chest roentgenogram of an infant demonstrating that the heart (H) is located in the right chest (dextrocardia), and there is left to right reversal of viscera (situs inversus). In this condition, the liver (Liv) is on the left side of the abdomen while the stomach (Sto) is on the right, i.e., situs inversus totalis. Tr, trachea. Reproduced from reference [4].

**Figure 9 children-09-01977-f009:**
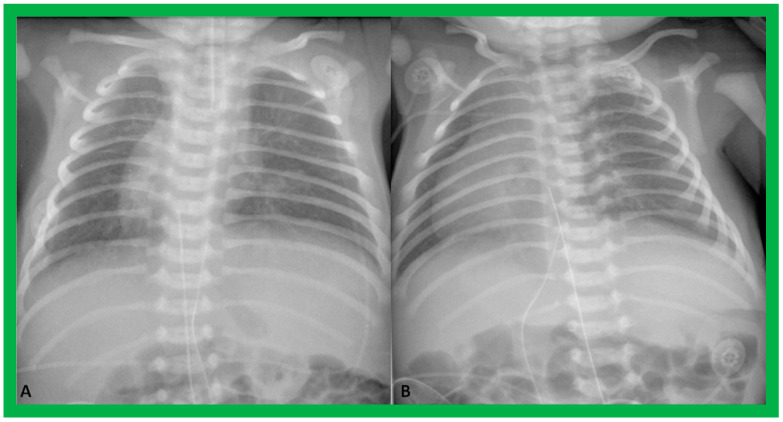
Chest roentgenograms of two babies with dextrocardia demonstrating liver across the entire abdomen. These babies were later diagnosed with asplenia syndrome. Umbilical venous and arterial catheters (not labeled) are seen in both (**A**) and (**B**).

**Figure 10 children-09-01977-f010:**
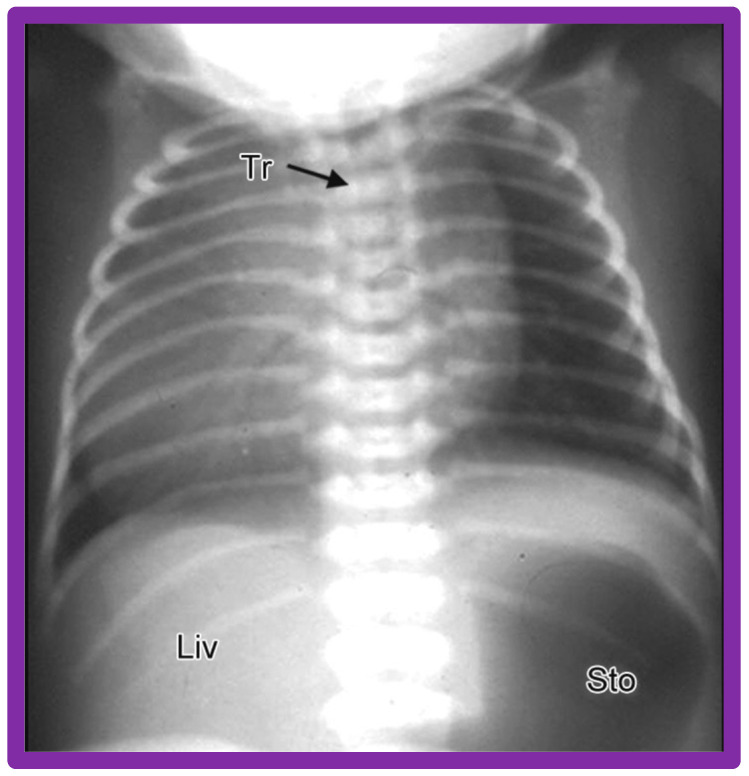
A chest roentgenogram of a baby illustrating dextrocardia along with normal visceral position (situs solitus) with the liver (Liv) on the right side and the stomach (Sto) on the left side, i.e., isolated dextrocardia. The tracheal (Tr) position in the middle is shown. The Sto is dilated, probably related to intestinal obstruction secondary to malrotation of the gut. Modified from reference [4].

**Figure 11 children-09-01977-f011:**
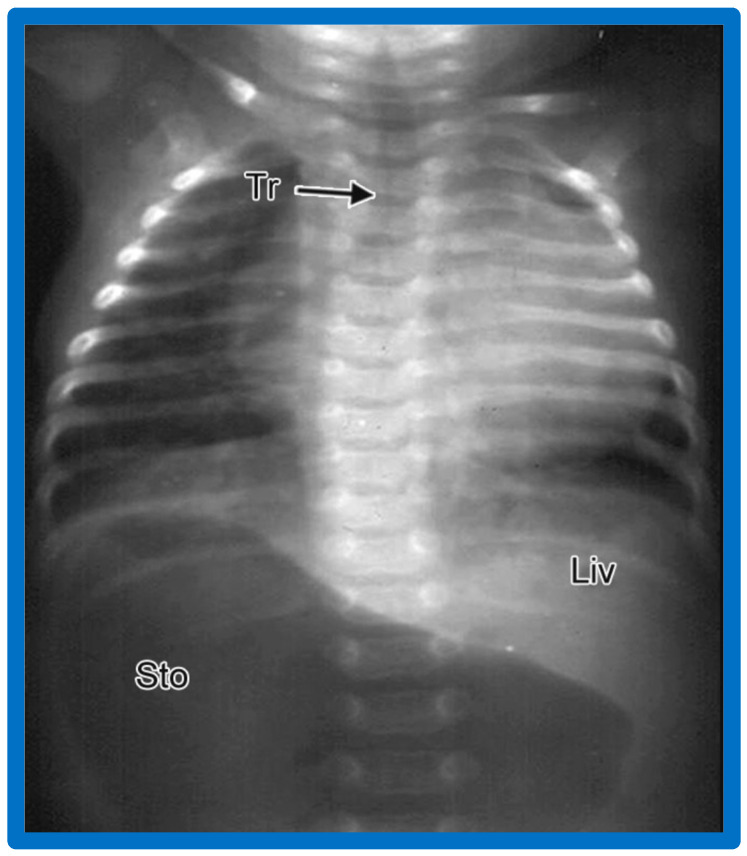
A chest roentgenogram of a baby who has isolated levocardia. The heart is on the left side, which is normal (levocardia), while the stomach (Sto) and the liver (Liv) are reversed (situs inversus). The marked dilatation of the Sto is likely due to intestinal obstruction secondary to malrotation of the gut. Modified from reference [4].

**Figure 12 children-09-01977-f012:**
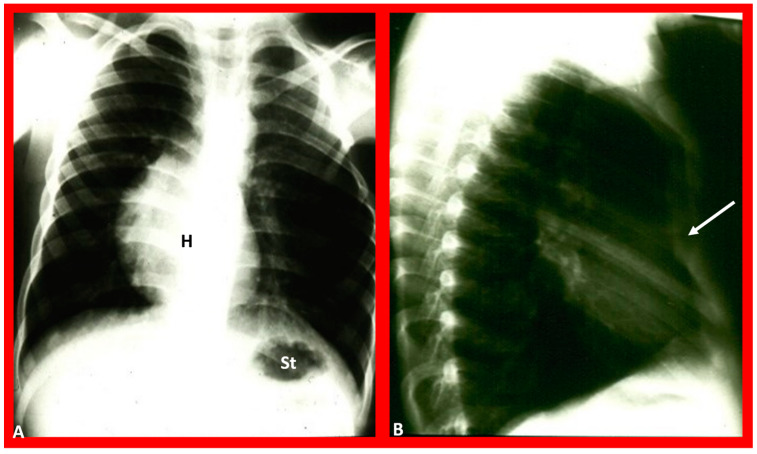
Chest X-rays of a five-year-old patient who was seen for evaluation since the referring physician noted a right-sided heart. Physical examination was unremarkable. (**A**) The heart (H) is displaced to the right, and there was no obvious pulmonary pathology. The gaseous shadow of the stomach (St) is located on the left side (normal). (**B**) The sternum (arrow) is showing posterior displacement, presumably causing dextroposition of the heart. Modified from reference [3].

**Figure 13 children-09-01977-f013:**
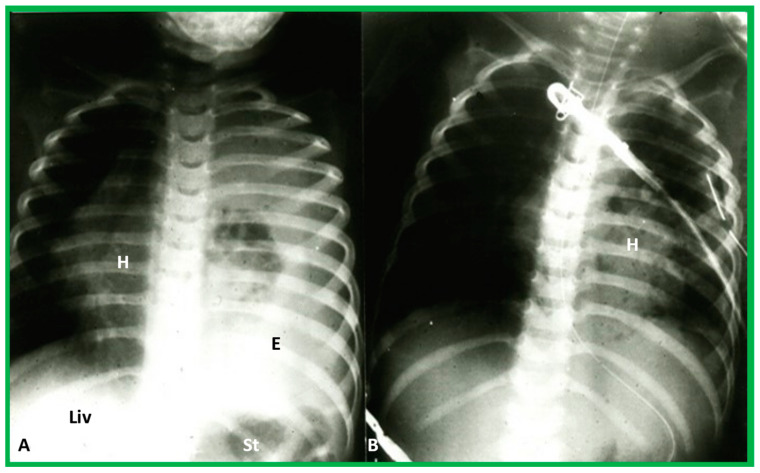
(**A**) Chest X-ray of a child with left-sided empyema (E), displacing the heart (H) to the right side. The liver (Liv) and stomach bubble (St) are in a normal position, indicating no evidence of situs abnormality. (**B**) Following drainage of empyema, the H is moved back into the left side of the chest. Modified from reference [3].

**Figure 14 children-09-01977-f014:**
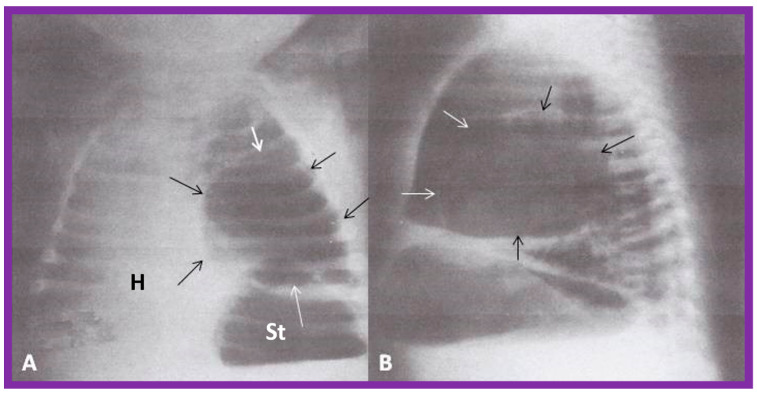
Chest film in posteroanterior (**A**) and lateral (**B**) projections demonstrate a large pulmonary cyst, shown with arrows. As one can see, the heart (H) is pushed to the right, i.e., dextroposition of the heart. St, stomach bubble. Reproduced from reference [22].

**Figure 15 children-09-01977-f015:**
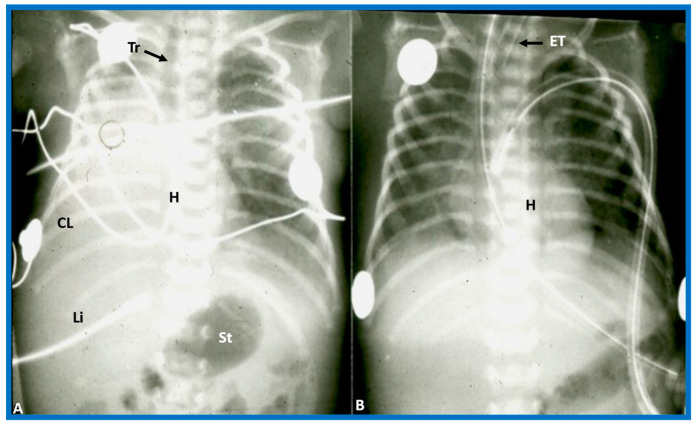
(**A**) Chest X-ray of a baby who had collapsed right lung (CL), pulling the heart (H) to the right, causing dextroposition of the H. The trachea (Tr) is shifted to the right side. The liver (Li) and stomach (St) bubble are in the normal position (situs solitus). (**B**) Following endotracheal (ET) intubation and suctioning, the right lung expanded, returning the H to its normal position. Modified from reference [4].

**Figure 16 children-09-01977-f016:**
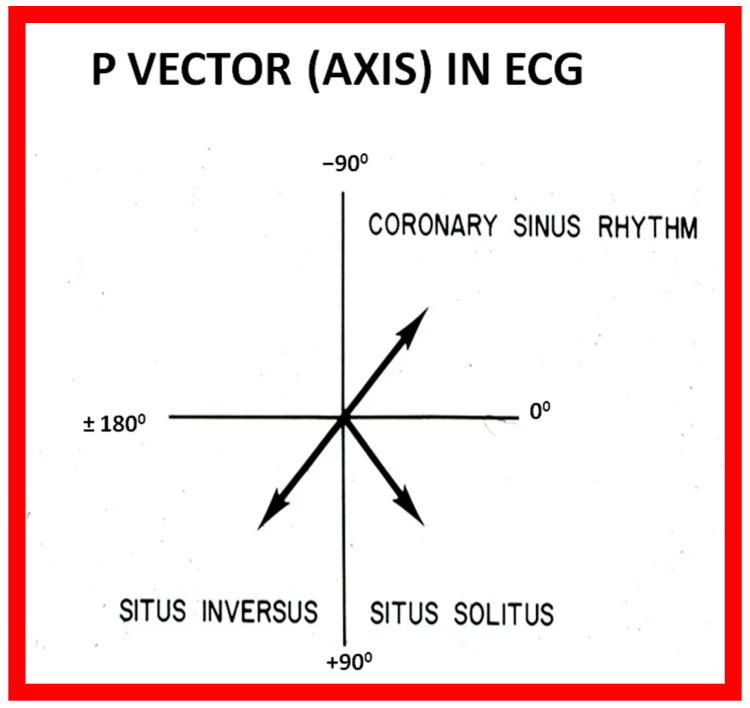
Diagram illustrating the utility of frontal plane P wave vector (axis). Axis of the P wave in-between 0° and +90° indicates normally positioned atria (situs solitus of the atria). Axis of the P wave in-between +90° and ±180° is suggestive of reversed atrial position (situs inversus of the atria). Axis of the P wave in-between 0 and −90° is termed low atrial or coronary sinus rhythm; such P waves are not useful in assessing the situs of the atria. Reproduced from reference [2].

**Figure 17 children-09-01977-f017:**
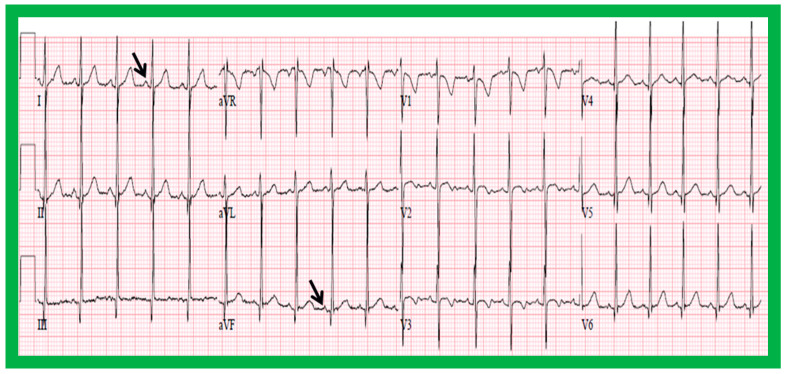
Electrocardiogram (ECG) of a patient illustrating normal P waves, namely, positive P waves in both leads I and AVF (axis of +45 degrees), which suggests normal atrial position; situs solitus. The ECG also shows normal Q waves in both leads V5 and V6 and no Q waves in leads in both V1 and V2; these data suggest a normal relationship of ventricles with the right ventricle on the right side and left ventricle on the left side. Reproduced from reference [4].

**Figure 18 children-09-01977-f018:**
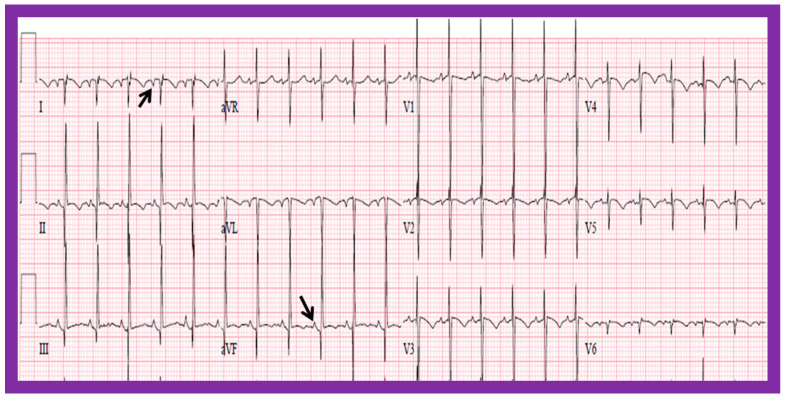
Electrocardiogram (ECG) of a patient illustrating an abnormal vector of the P wave (+135 degrees). The P waves are negative in lead I, while they are positive in lead AVF (arrows). Such a pattern is indicative of situs inversus with a morphologic left atrium on the right side and a morphologic right atrium on the left side. The ECG also shows no Q waves in the QRS complexes of the chest leads; such a pattern is not helpful in localizing the ventricles. Reproduced from reference [4].

**Figure 19 children-09-01977-f019:**
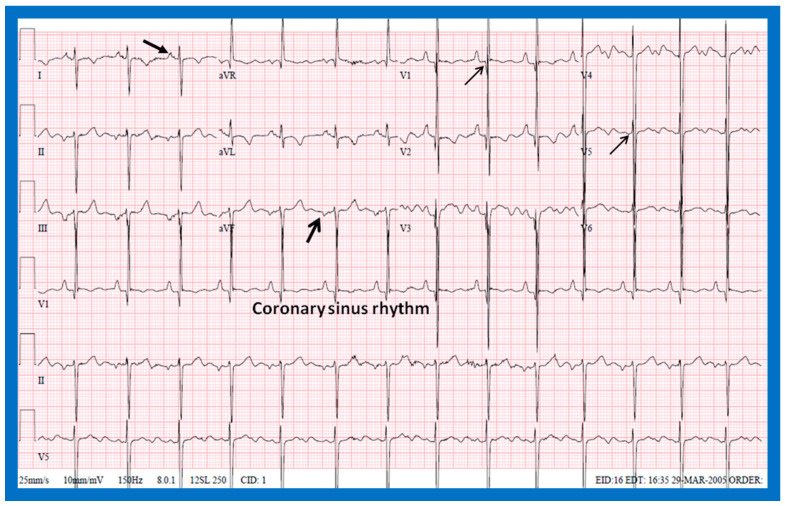
Electrocardiogram (ECG) of a patient illustrating positive P waves in lead I and negative P waves in lead AVF (thick arrows) with a P wave vector of −45 degrees. Such a pattern is suggestive of a low atrial or coronary sinus rhythm. However, such P wave vectors are not useful in situs localization. The ECG also shows Q waves in the right chest leads (V1 and V2) and no Q waves in the left chest leads (V5 and V6) (thin arrows); such a pattern is indicative that the ventricles are inverted. However, such an abnormality may also be observed in patients with right ventricular hypertrophy of a severe degree. Reproduced from reference [4].

**Figure 20 children-09-01977-f020:**
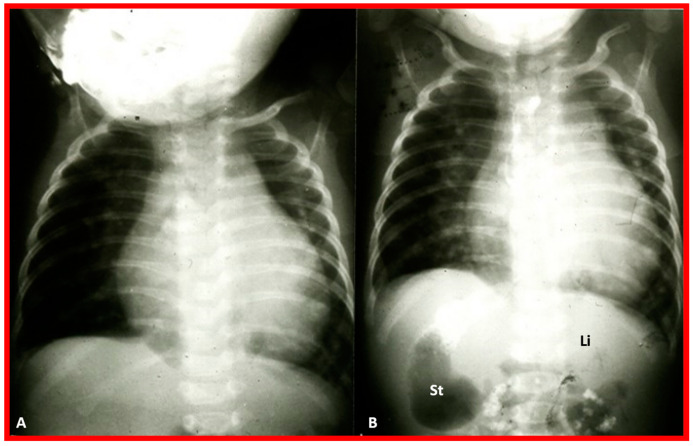
Value of stomach (St) gas shadow in determining situs localization is shown in this figure. In (**A**), the St gas bubble was not seen, and therefore, visceral situs could not be classified. In a subsequent chest X-ray (**B**), the St gas was seen, indicating situs inversus. This case exemplifies the requirement for seeing the St gaseous opacity on the chest X-ray. Li, liver. Modified from reference [3].

**Figure 21 children-09-01977-f021:**
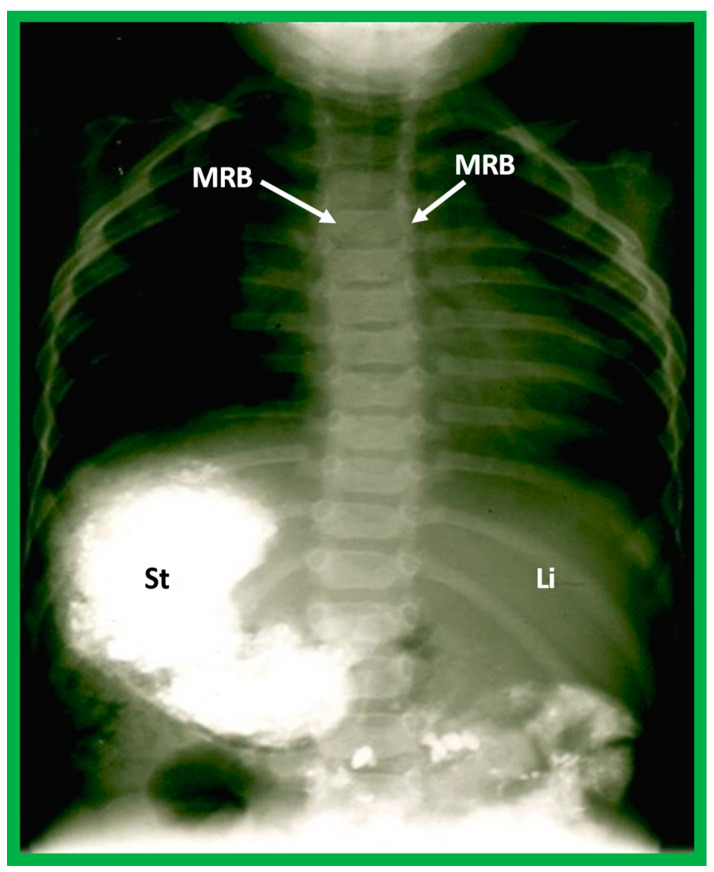
Barium study showing barium in the stomach (St) on the right side while the liver (Li) is seen on the left, indicating situs inversus. The heart is on the left side, indicating isolated levocardia. Tracheobronchial tree pattern shows morphologic right bronchus (MRB) on both sides, indicative of dextro-isomerism.

**Figure 22 children-09-01977-f022:**
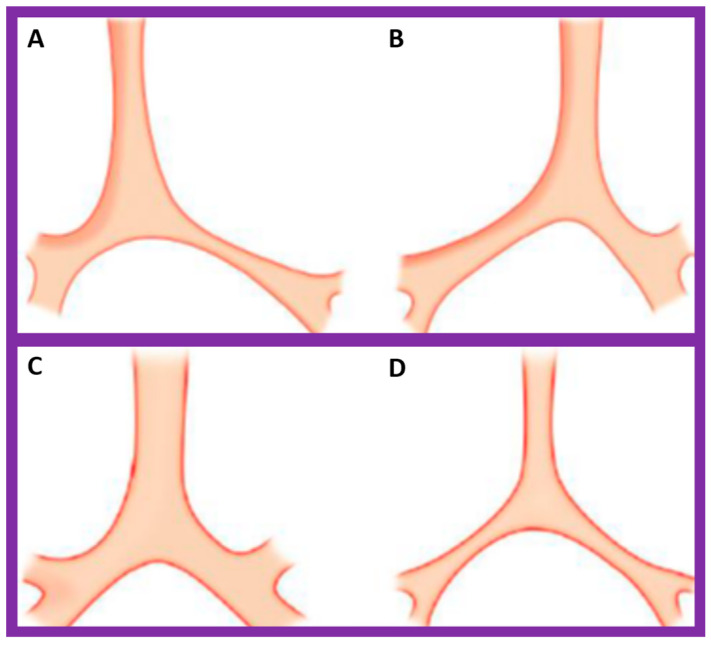
Diagrammatic depiction of the anatomy of the tracheobronchial tree. (**A**) In patients with situs solitus, the right bronchus is short and wide and inclines somewhat steeply, whereas the left bronchus is long and narrow and traverses in a horizontal pattern. (**B**) In patients with situs inversus, the configuration of the bronchial pattern is inverted; the morphologic right bronchus is on the left side, while the morphologic left bronchus is on the right side. (**C**) In patients with heterotaxy (asplenia syndrome), both right and left bronchi have the morphology of the right bronchus. (**D**) In subjects with heterotaxy (polysplenia syndrome), both the right and left bronchi have the appearance of morphologic left bronchi. Concept is derived from Reference [2], and the Figure is modified from Reference [5].

**Figure 23 children-09-01977-f023:**
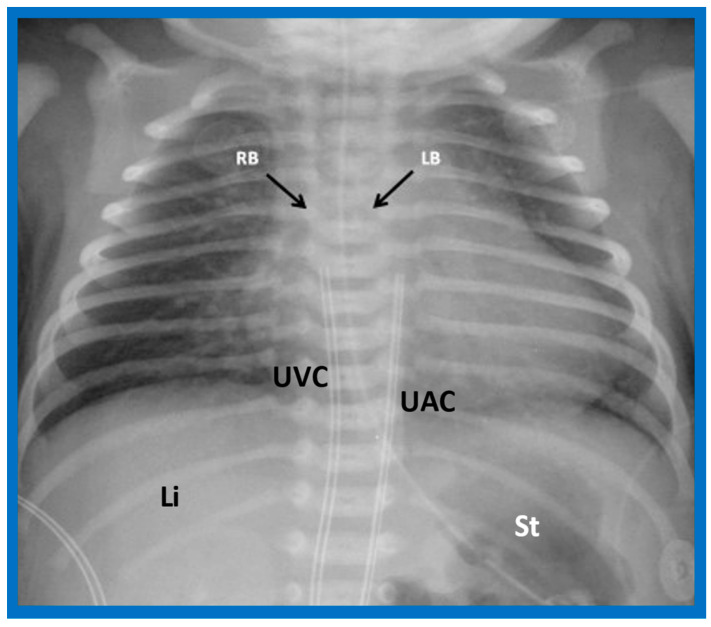
A chest roentgenogram of a patient with situs solitus demonstrating a wide and short right bronchus (RB) which descends somewhat steeply, whereas the left-sided bronchus (LB) is narrow and long and traverses horizontally. The liver (Li) is seen on the right side, and the stomach (St) bubble is seen on the left side, consistent with situs solitus. Mild cardiac enlargement and increased lung vascular patterns are also seen. UAC indicates an umbilical arterial catheter, and UVC points to the catheter in the umbilical vein. Modified from reference [4].

**Figure 24 children-09-01977-f024:**
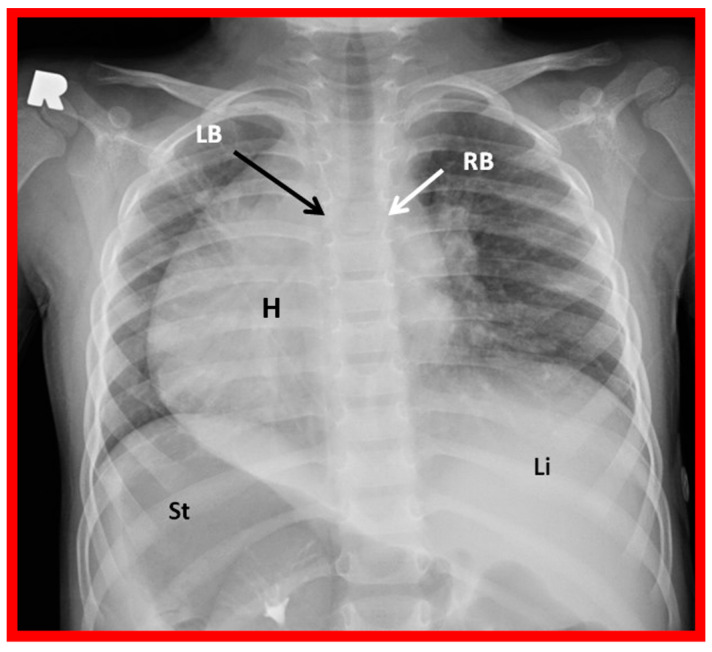
A chest roentgenogram of a patient with situs inversus demonstrating left-to-right reversal of the normal bronchial pattern; the morphologically right bronchus (RB) is on the left side, and the morphologically left bronchus (LB) is on the right side. The left-to-right inversion of the viscera with the liver (Li) on the left and the stomach (St) on the right indicates situs inversus. The heart (H) is in the right hemithorax (dextrocardia). Modified from reference [4].

**Figure 25 children-09-01977-f025:**
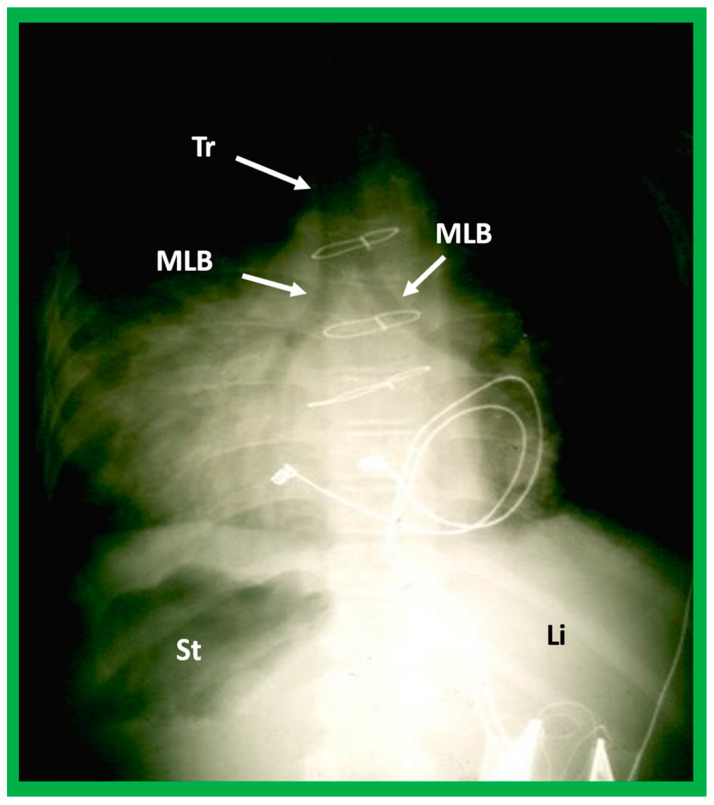
A highly penetrated chest roentgenogram in an anteroposterior view of a child with dextrocardia associated with situs inversus demonstrates bilateral morphologic left bronchi (MLB); the patient was found to have polysplenia syndrome. Sternal wires and pacemaker artifacts (not labeled) from the previous operation are also seen. Li, liver, St, stomach; Tr, trachea.

**Figure 26 children-09-01977-f026:**
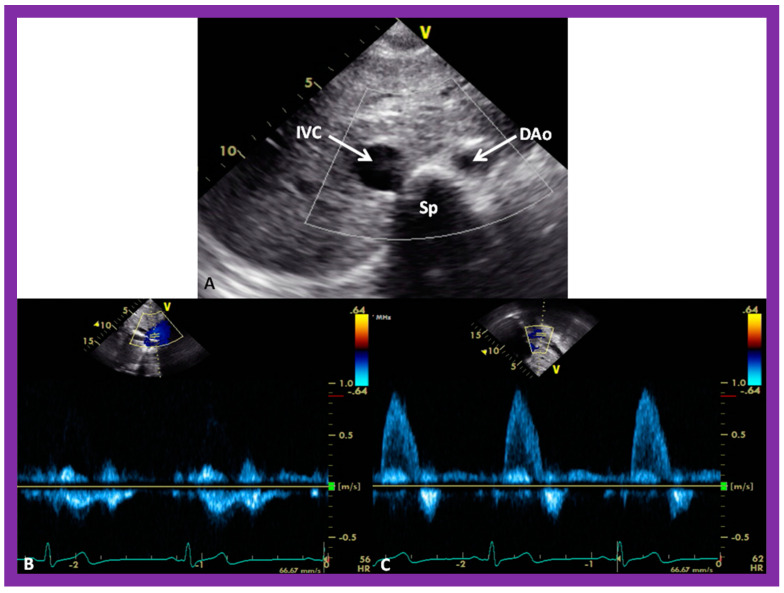
Selected video images secured in a short axis echo view of a normal child, illustrating the positions of the aorta (Ao) on the left side of the spine (Sp) and the inferior vena cava (IVC) on the right side of the Sp. (**A**) The IVC is generally larger than the Ao. (**B**,**C**) demonstrate Doppler flow characteristics with low-velocity venous flow in the IVC (**B**) and a higher velocity arterial flow in the Ao (**C**). These data are indicative of situs solitus. Reproduced from reference [4].

**Figure 27 children-09-01977-f027:**
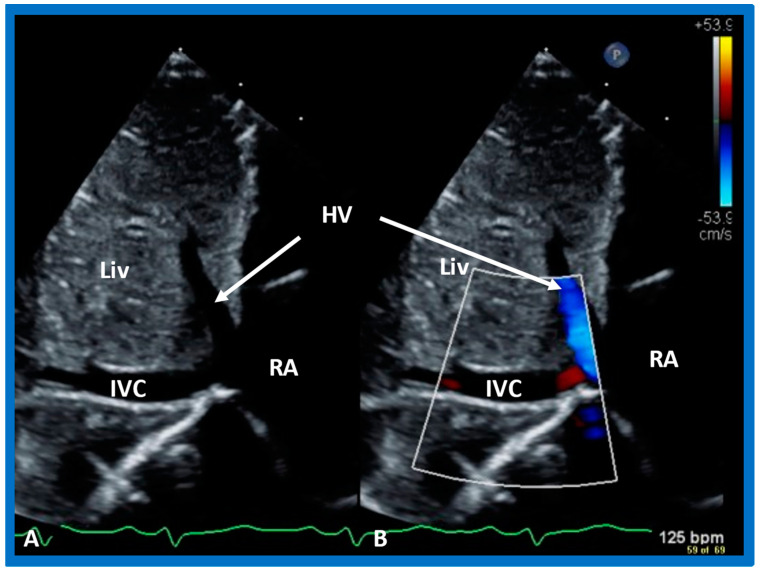
Subcostal echo images in two-dimensional (**A**) and color flow (**B**) illustrate the entrance of the inferior vena cava (IVC) into the right atrium (RA). HV, hepatic vein; Liv, liver.

**Figure 28 children-09-01977-f028:**
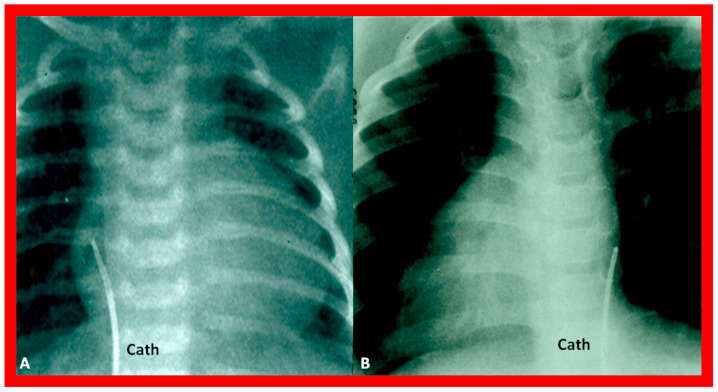
Cinefluoro images in posteroanterior views, illustrating positions of the catheters (Cath) in the inferior vena cava as they enter the right atrium (not labeled) in a patient with situs solitus (**A**) and another patient with situs inversus (**B**). Reproduced from reference [4].

**Figure 29 children-09-01977-f029:**
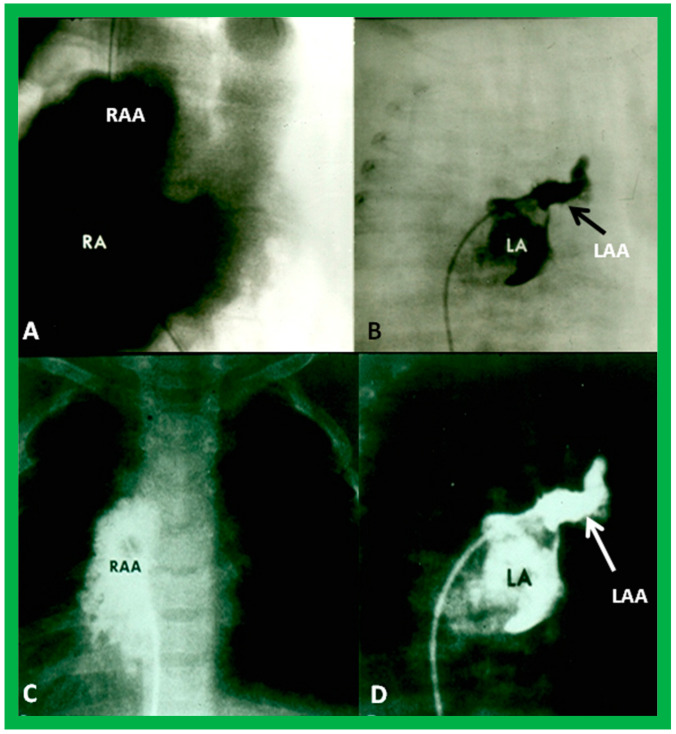
Angiographic images of right (RA) (**A**,**C**) and left (LA) (**B**,**D**) atria in a posteroanterior view illustrating the atrial appendage morphology. The RA appendage (RAA) has a broad, large, and pyramidal shape (**A**,**C**), whereas the LA appendage (LAA) has a narrow, small, and tubular shape (**B**,**D**). Reproduced from reference [4].

**Figure 30 children-09-01977-f030:**
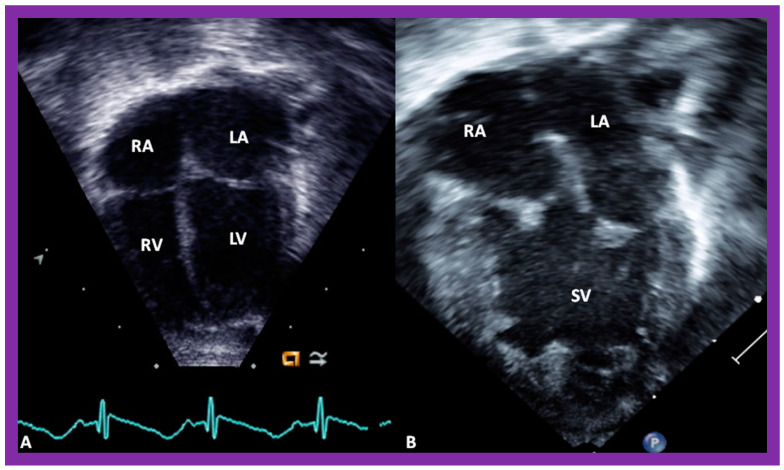
Selected pictures from apical four-chamber echo views of two children, the first (**A**), who has two ventricles, and the second (**B**), who has one (single) ventricle (SV). The left atrium (LA), left ventricle (LV), right atrium (RA), and right ventricle (RV) are labeled. Reproduced from reference [4].

**Figure 31 children-09-01977-f031:**
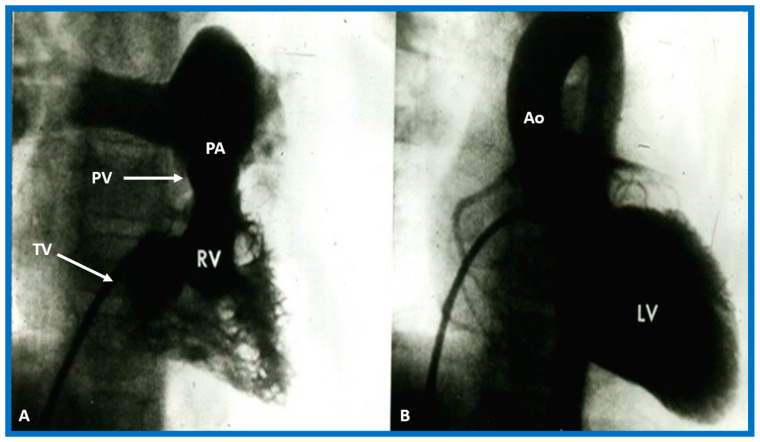
Selected angiographic images demonstrating characteristic features of the right (**A**) and left (**B**) ventricles. The right ventricle (RV) exhibits coarse trabeculations (**A**), while the left ventricle (LV) shows smooth or fine trabeculations (**B**). In addition, the RV exhibits a triangular shape, whereas the LV is a foot-shaped structure. The tricuspid (TV) and pulmonary (PV) valves (arrows in (**A**)) are separated by a muscular structure, crista supraventricularis. Aorta (Ao) and pulmonary artery (PA) are labeled. Modified from reference [3].

**Figure 32 children-09-01977-f032:**
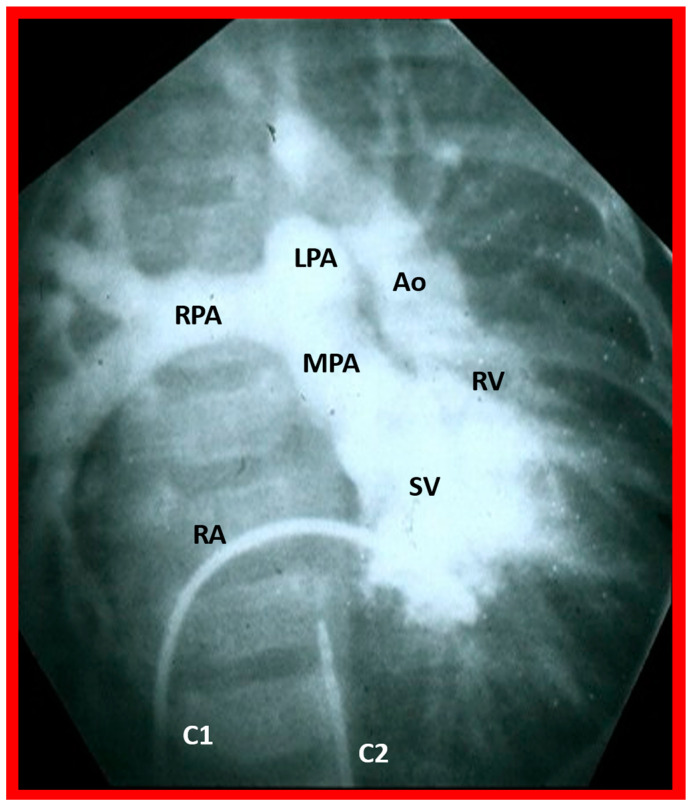
Selected angiographic image in the posteroanterior view of a patient with a single ventricle (SV) illustrating opacification of the main pulmonary artery (MPA) from the main ventricular chamber and the aorta (Ao) arising from a smallish right ventricle (RV). The Ao is situated to the left of the pulmonary artery, signifying the l-transposition of the great arteries. C1. The catheter in the unmarked inferior vena cava was advanced into the right atrium (RA) and then manipulated into the ventricle; C2. Catheter in the unmarked descending aorta. The left pulmonary artery (LPA) and right pulmonary artery (RPA) are labeled. Reproduced from reference [31].

**Figure 33 children-09-01977-f033:**
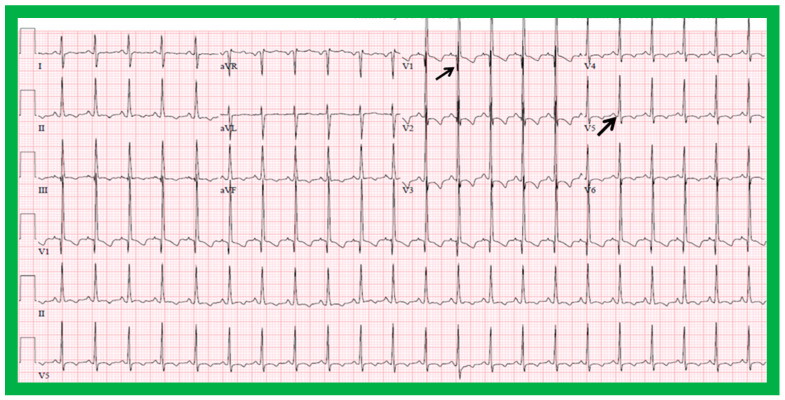
Electrocardiogram of a child illustrating q waves in lead V1 (top arrow) and no q waves in leads V5 and V6 (lower arrow). Such a pattern is indicative of inversion of the ventricles. However, such an abnormality has also been seen in subjects with severe right ventricular hypertrophy. A P wave vector of +45 degrees with positive P waves in leads I and AVF is indicative of situs solitus.

**Figure 34 children-09-01977-f034:**
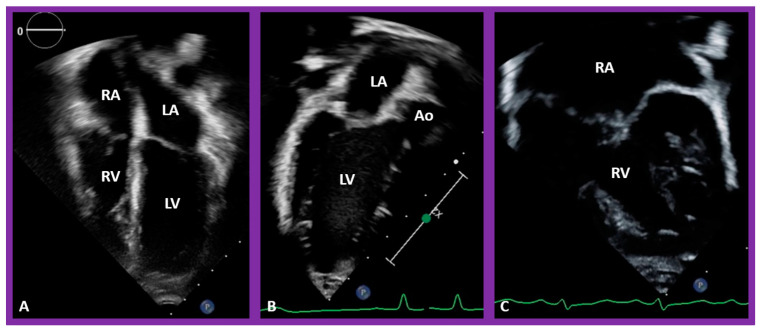
Echo images demonstrating finely trabeculated left ventricle (LV) (**A**,**B**) and coarsely trabeculated right ventricle (RV) (**A**,**C**) in apical precordial views. The LV trabeculations are barely seen in (**A**) but are better seen in (**B**). Similarly, the RV trabeculations are better seen in (**C**). Aorta (Ao), left atrium (LA), and right atrium (RA) are labeled.

**Figure 35 children-09-01977-f035:**
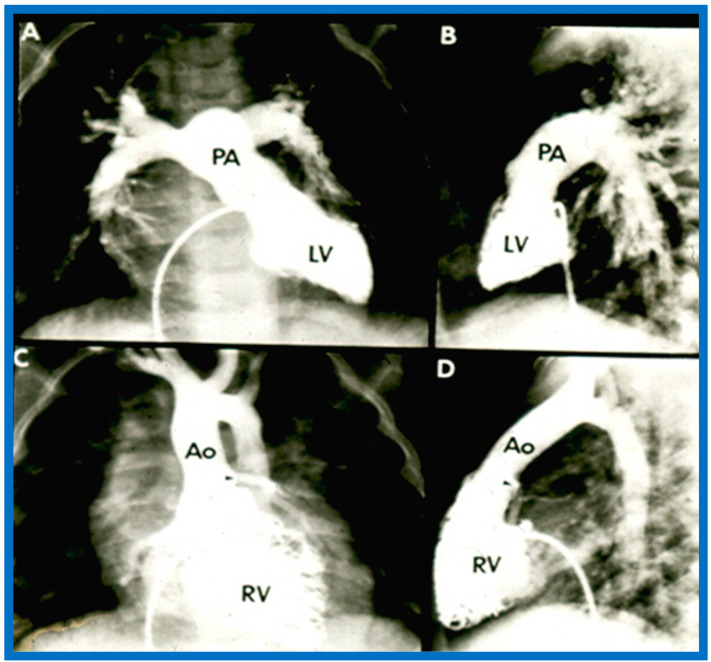
(**A**,**B**). Selected cine frames of the Left ventricle (LV) in posteroanterior (**A**) and lateral (**B**) projections of a child with d-transposition of the great arteries illustrating morphologic LV with fine trabeculations. The pulmonary artery (PA) originates from the LV. The pulmonary valve is positioned inferior and posterior to its normal position. (**C**,**D**). Selected cine frames of the right ventricle (RV) in posteroanterior (**C**) and lateral (**D**) projections of the same child shown in (**A**,**B**) illustrating morphologic RV with coarse trabeculations. The aorta (Ao) originates from the RV. The aortic valve is positioned superior and anterior (**C**,**D**) to its normal position. The closeness of the mitral with the pulmonary valve in a morphologic LV (**A**,**B**) and the separation of tricuspid and aortic valves in a morphologic RV (particularly in **D**) is clearly seen. Replicated from reference [35].

**Figure 36 children-09-01977-f036:**
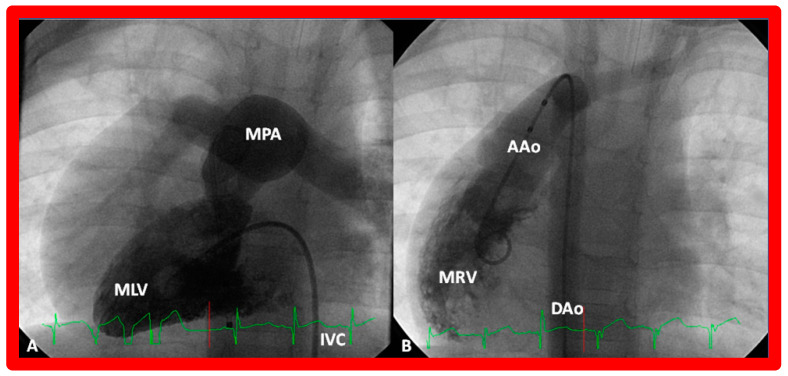
(**A**) Cine frame from morphologic left ventricular (MLV) cineangiogram in a posteroanterior view of an infant with dextrocardia, exhibiting fine trabeculations of the MLV on the left side. The MLV gives origin to the main pulmonary artery (MPA), which is dilated. The position of the catheter in the inferior vena cava (IVC) is on the left side of the spine. (**B**) Demonstrates a morphologic right ventricle (MRV) in a posteroanterior view in the infant shown in (**A**); coarse trabeculations are seen. The right-sided MRV gives rise to the aorta, denoted by AAo. The aorta descends on the right side of the spine. The aortic valve is situated rightward and superior to the pulmonary valve. It is located anteriorly in the lateral view (not shown). This information suggests a d-loop of the ventricles and d-transposition of the great arteries in a subject with dextrocardia, a disorder suggestive of corrected transposition physiology. DAo, descending aorta. Replicated from reference [36].

**Figure 37 children-09-01977-f037:**
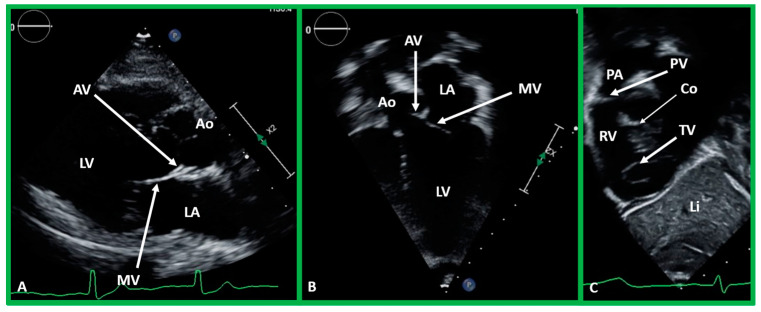
(**A**,**B**). Echo images in the parasternal long axis (**A**) and modified apical four-chamber (**B**) views to demonstrate the atrioventricular valve-to-semilunar valve relationship. Note that the aortic (AV) and mitral (MV) valve leaflets are in fibrous continuity with each other, indicating that this ventricle is a morphologic left ventricle (LV). (**C**) Selected video frame in a subcostal view to demonstrate atrioventricular valve-to-semilunar valve relationship in a child with normally related great vessels demonstrating lack of continuity between the pulmonary (PV) and tricuspid (TV) valve leaflets due to conus (Co) separating them. Aorta (Ao), left atrium (LA), liver (Li), pulmonary artery (PA), and right ventricle (RV) are labeled.

**Figure 38 children-09-01977-f038:**
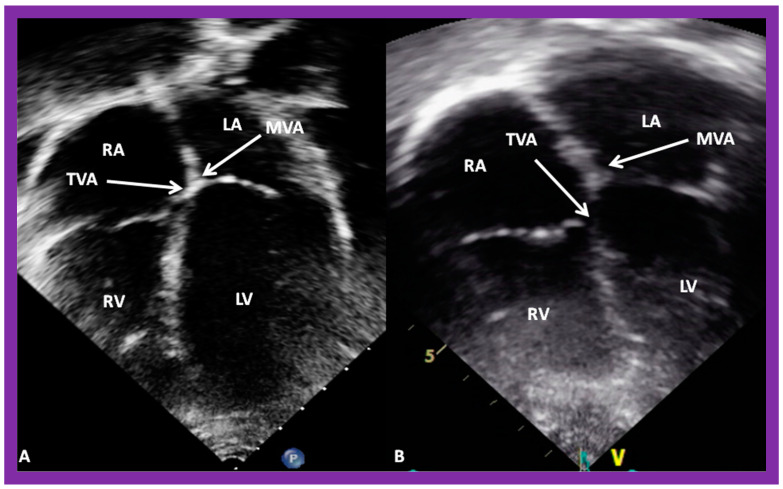
(**A**,**B**). Echo images in apical four-chamber views of 2 children with a normal interventricular relationship. A superior level of mitral valve attachment (MVA) relative to attachment of the tricuspid valve (TVA) is seen in both children. Left atrium (LA), left ventricle (LV), right atrium (RA), and right ventricle RV are labeled. Replicated from reference [4].

**Figure 39 children-09-01977-f039:**
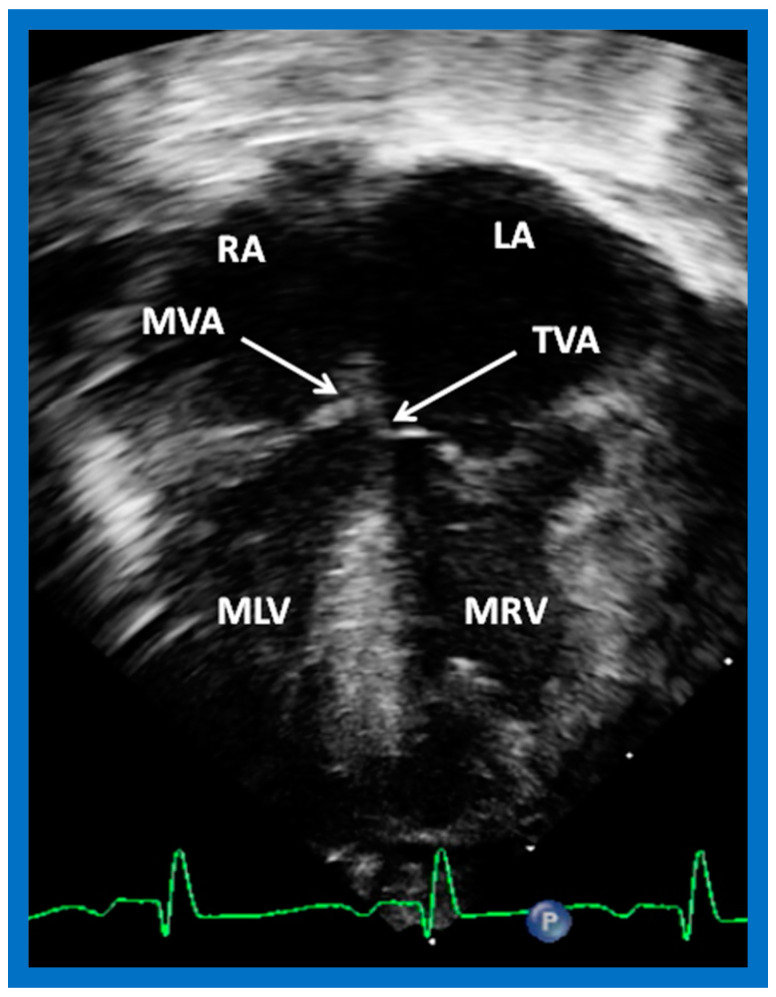
Selected echo image in the apical four-chamber projection of a child with ventricular inversion. Note the superior level of mitral valve attachment (MVA) on the right side relative to the attachment of the tricuspid valve (TVA) on the left side, signifying that the ventricles are inverted. This is a reversal of what is illustrated in Figure 38. Left atrium (LA), morphologic left ventricle (MLV), morphologic right ventricle (MRV), and right atrium (RA) are labeled. Replicated from reference [4].

**Figure 40 children-09-01977-f040:**
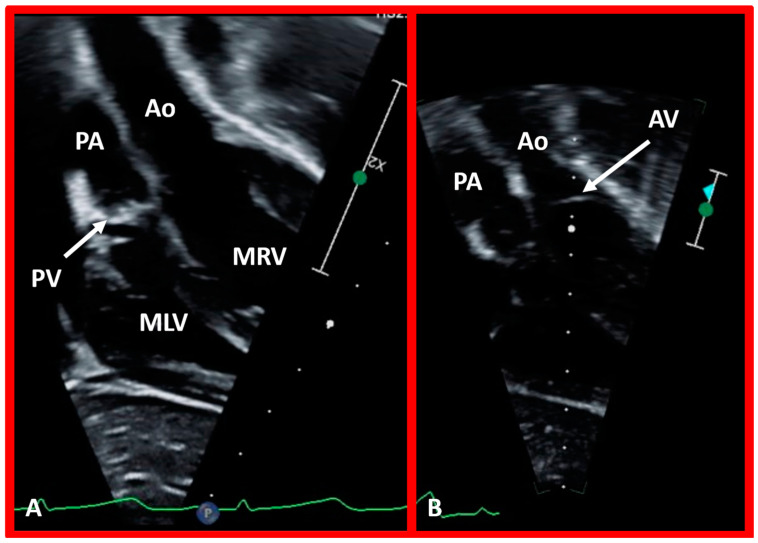
Echo images in apical four-chamber projections of a child with ventricular inversion. The aortic valve (AV) and aorta (Ao) are located to the left of the pulmonary valve (PV) and pulmonary artery (PA), indicating an l-loop, and that morphologic right ventricle (MRV) is on the left, and the morphologic left ventricle (MLV) is on the right. In (**A**), the PV (arrow in (**A**)) is clearly seen, while AV is imaged in (**B**) (arrow in (**B**)).

**Figure 41 children-09-01977-f041:**
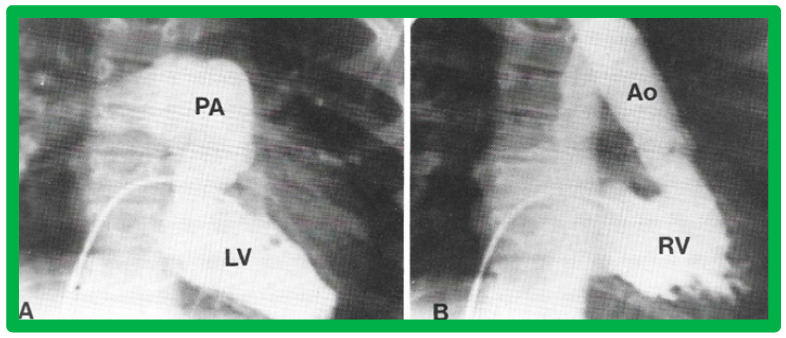
Cine images demonstrating an l-loop with the aortic valve located to the left of the pulmonary valve in a baby with corrected transposition of the great vessels. The aortic valve is also superior (**B**) and anterior (in lateral view—not shown) of the pulmonary valve. (**A**) Smooth-walled morphologic left ventricle (LV) is on the right side, giving origin to the pulmonary artery (PA). (**B**) Left-sided, morphologic right ventricle (RV) is coarsely trabeculated and gives origin to the aorta (Ao). The angiogram in (**B**) was obtained via a catheter positioned from the inferior vena cava to the right atrium, then into the left atrium through a patent foramen ovale, and then into the morphologic RV. Replicated from reference [39].

**Figure 42 children-09-01977-f042:**
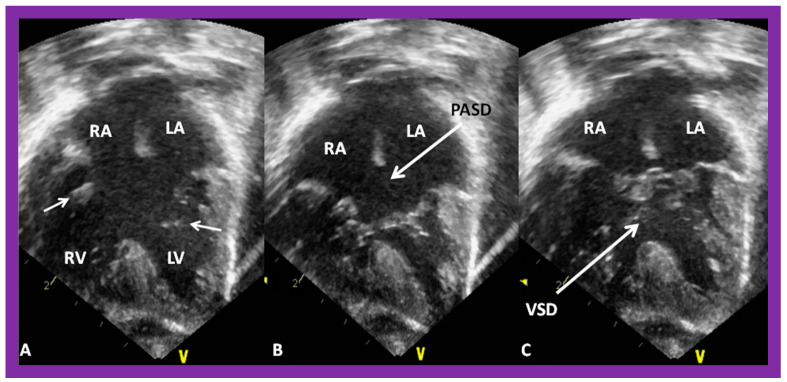
Echo images in apical four-chamber projections demonstrating the connection of a common atrioventricular valve with the right (RV) and left (LV) ventricles. In (**A**) (small arrows), the atrioventricular (AV) valve is open, whereas the AV valve is closed in (**B**,**C**) during varying phases of the cardiac cycle. Atrial septal defect of ostium primum (PASD) type of large size in (**B**), as well as a large ventricular septal defect (VSD) in (**C**), are illustrated. LA, left atrium; LV, left ventricle; RA, right atrium; RV, right ventricle. Replicated from reference [41].

**Figure 43 children-09-01977-f043:**
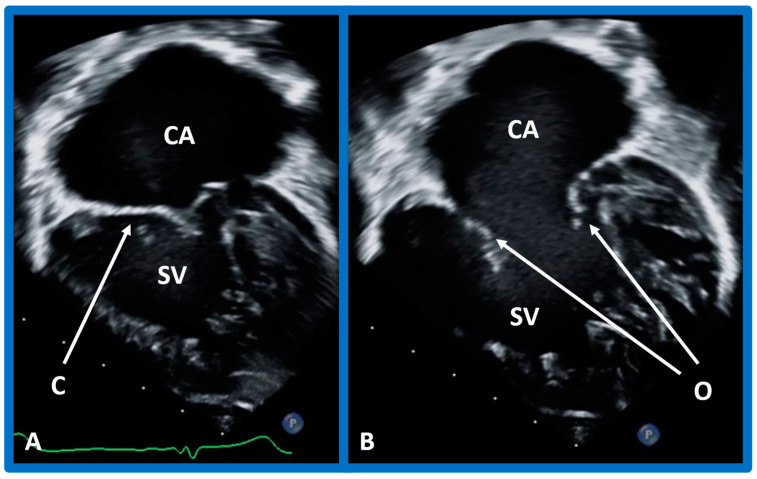
Echo frames in apical four-chamber views demonstrating a connection of common atrium (CA) with single ventricle (SV) by a one (common) atrioventricular valve. In (**A**), the common atrioventricular valve is closed (C), whereas in (**B**), it is open (O). Heavy trabeculation is seen in the SV.

**Figure 44 children-09-01977-f044:**
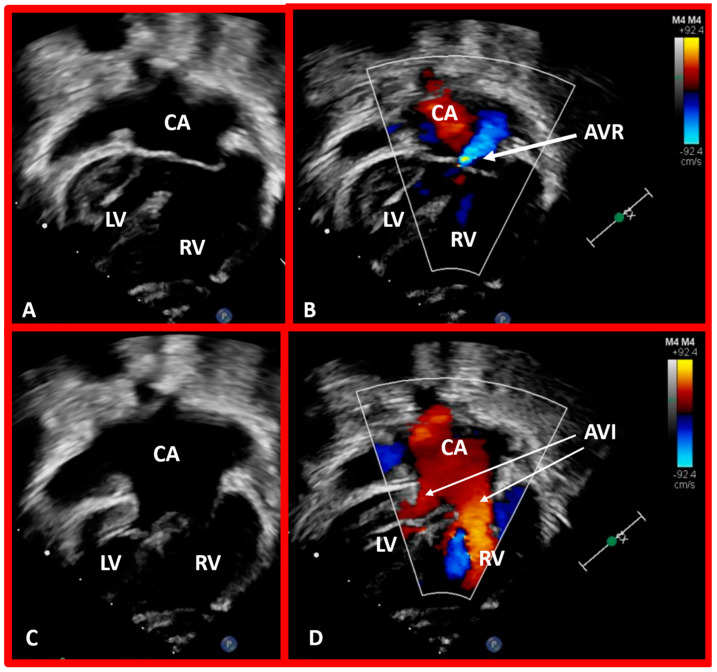
Selected echo frames in apical four-chamber views of a patient with dextrocardia and ventricular inversion demonstrating the connection of the common atrium (CA) with both ventricles via a common atrioventricular valve. The common atrioventricular valve is completely closed in (**A**,**B**), whereas it is open in (**C**,**D**). Atrioventricular valve regurgitation (AVR) is shown in (**B**) with a thick arrow. Atrioventricular inflow (AVI) is marked with thin arrows in (**D**). Left ventricle (LV) and right ventricle (RV) are labeled.

**Figure 45 children-09-01977-f045:**
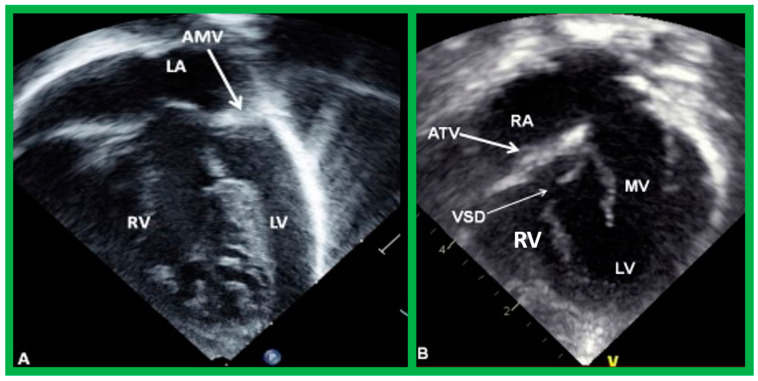
(**A**) Echo frames in apical four-chamber view of a patient with mitral atresia demonstrating atretic mitral valve (AMV), pointed out by an arrow in (**A**). A large right ventricle (RV), a small left atrium (LA), and a small left ventricle (LV) are also visualized. (**B**) Echo frames in apical four-chamber view of a patient with tricuspid atresia, demonstrating atretic tricuspid valve (ATV) shown by the thick arrow in (**B**). Dilated LV, a small RV, and moderate-sized ventricular septal defect (VSD; thin arrow in (**B**)) are also seen. Mitral valve (MV) and right atrium (RA) are labeled.

**Figure 46 children-09-01977-f046:**
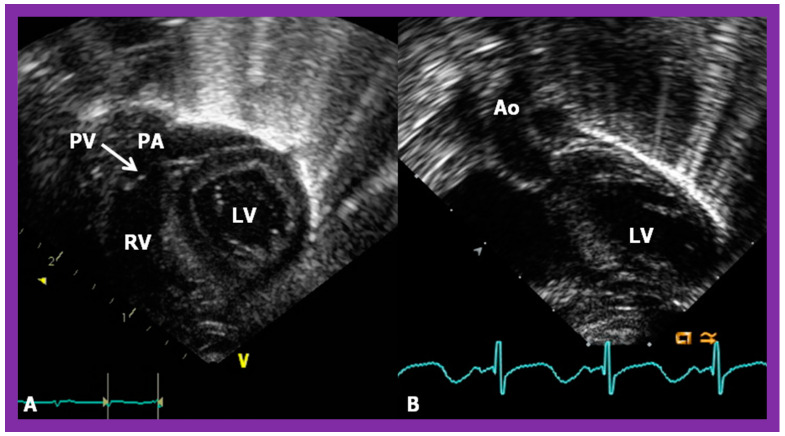
Echo images in subcostal long-axis projections of the right (RV) (**A**) and left (LV) (**B**) ventricles illustrating the position of the pulmonary valve (PV). The PV is located higher and anterior to the aortic valve (AV). The AV is located on the right side of the PV in straight posteroanterior view (not shown in this illustration). As expected, the LV is seen posteriorly while the RV is anterior. Aorta (Ao) and pulmonary artery (PA) are labeled. Replicated from reference [41].

**Figure 47 children-09-01977-f047:**
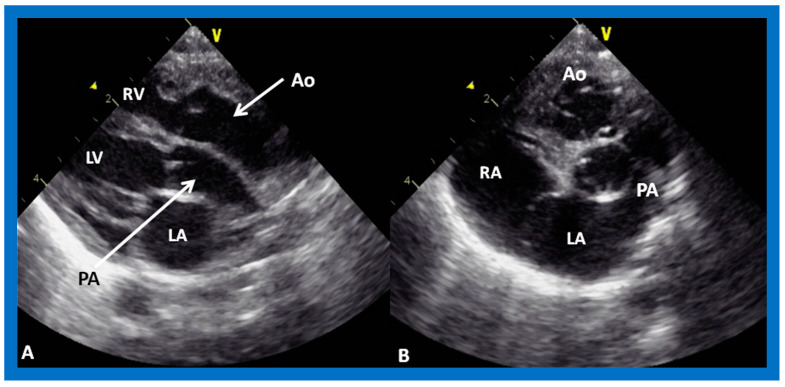
Echo images in parasternal long (**A**) and short (**B**) axis projections of a baby with transposition of the great vessels illustrating anterior (**A**,**B**) and right-ward (**B**) position of the aortic valve and aorta (Ao) relative to the position of the pulmonary valve and pulmonary artery (PA). These images also demonstrate that the PA and Ao are in parallel position (**A**). Left atrium (LA), left ventricle (LV), right atrium (RA), and right ventricle (RV) are labeled. Replicated from reference [42].

**Figure 48 children-09-01977-f048:**
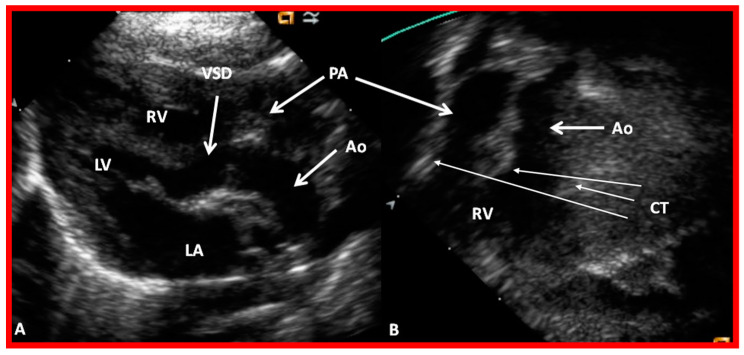
Echo frames from the parasternal long axis (**A**) and subcostal (**B**) views illustrating that both the aorta (Ao) and the pulmonary artery (PA) arise from the right ventricle (RV), i.e., double-outlet right ventricle. Note that the great vessels are normally related to each other. A ventricular septal defect (VSD) (vertical arrow) is also seen. The conal tissue (CT) is shown by thin arrows in (**B**). The vessel marked PA was traced further and demonstrated to subdivide into right and left pulmonary arteries. Left atrium (LA) and left ventricle (LV) are labeled. Modified from reference [43].

**Figure 49 children-09-01977-f049:**
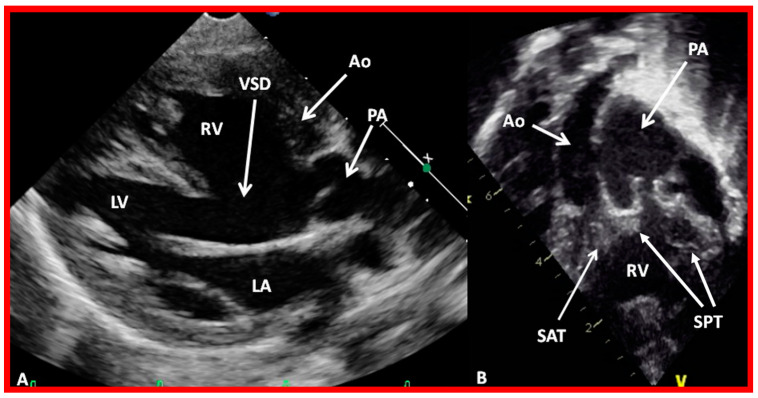
Echo images from the parasternal long axis (**A**) and subcostal (**B**) projections of a child with Taussig-Bing type of double-outlet right ventricle (with transposition of great vessels) illustrating that both the pulmonary artery (PA) and the aorta (Ao) arise from the right ventricle (RV). A large ventricular septal defect (VSD) and subaortic (SAT) and subpulmonary (SPT) tissues are also shown. The vessel marked PA was further traced and was demonstrated to divide into right and left PAs. Left atrium (LA) and left ventricle (LV) are labeled. Modified from reference [43].

**Figure 50 children-09-01977-f050:**
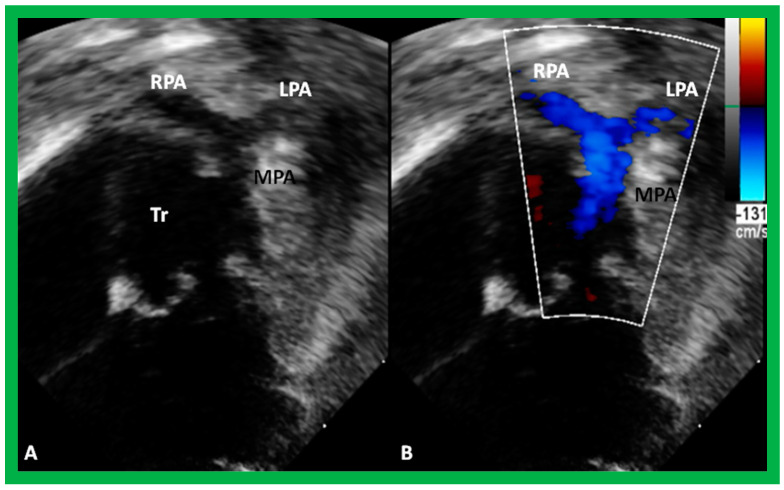
Selected echo frames from modified apical four-chamber views demonstrating the origin of one great vessel, truncus arteriosus (Tr), from the heart. Two dimensional (**A**) and color flow (**B**) frames demonstrating the origin of the main pulmonary artery (MPA) from the left of the Tr bifurcating into the right (RPA) and left (LPA) pulmonary arteries are shown. Replicated from reference [44].

**Figure 51 children-09-01977-f051:**
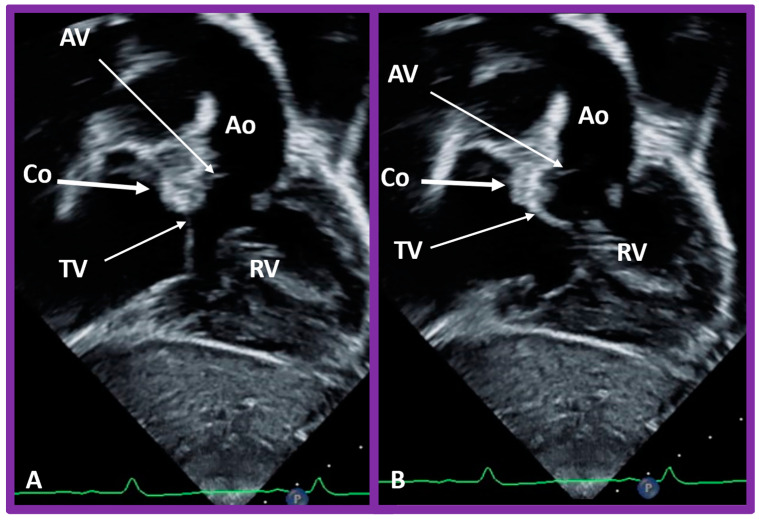
Selected video frames in subcostal views to demonstrate the atrioventricular valve-to-semilunar valve relationship in a child with l-transposition of the great vessels demonstrating a lack of continuity between the aortic valve (AV) and tricuspid valve (TV) leaflets due to conus (Co) separating them. The TV closed in (**A**) and open in (**B**). Ao, aorta; Right ventricle (RV).

## Data Availability

Not applicable.

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
