# Peer review of "Diagnosis of Dextrocardia with a Pictorial Rendition of Terminology and Diagnosis"

_children, 2022, doi:10.3390/children9121977_

Round 1

Reviewer 1 Report

The authors have written a very clear, thorough and well illustrated review of segmental analysis and a brief overview of management of various conditions associated with dextrocardia.  The senior author is particularly experienced and knowledgeable about segmental analysis, anomalies and dextrocardia conditions in particular.  

The strengths of this manuscript are the organized, clear writing and the well described process of determining the cardiac segmental anatomy for a precise and accurate diagnosis.  I think that the writing style makes this appealing and read-able for cardiologists and non-cardiologists alike.  The illustrations are excellent and plentiful which add a great deal of value to the content.  

As a cardiologist, I think the weakness of this manuscript is simply the very broad nature of the review.  I would suggest that the editors consider limiting this review to only the diagnosis of dextrocardia and a pictorial rendition of segmental analysis and terminology.  The article is quite long.  The section on management is not very specific to dextrocardia since the variety of physiology spans the entire spectrum of single ventricle and biventricular disease.  The authors have done an admirable job summarizing different strategies but the summary is essential the entire management of CHD.  I personally do not think the management section adds much and could be eliminated to keep the focus on the strong experience and knowledge of the authors on segmental analysis.  

Overall, I think this is a very strong review of segmental analysis and I applaud the authors on their clear, detailed and thorough writing of an inherently complicated topic. 

A few specific details and comments.  

- Figure 1 and 2 are redundant and one could be eliminated.

- 3.9 Isolated Dextrocardia – seems redundant with the Dextrocardia section.  Is this necessary?  Consider eliminating this subsection. 

- 3.10 Isolated Levocardia – also seems redundant and potentially confusing.  I would consider simply eliminating this sub-section.

- 3.11 Other Terms – given the length of this manuscript, may be worth cutting this section which doesn’t seem to add anything essential. 

- Line 168 – This is confusing to me.  How can the prevalence of dextrocardia be lower in an older age group and higher in neonates?  Dextrocardia isn’t a condition that changes over time, obviously.  Are the numbers implying some attrition (i.e. - death of neonates with dextrocardia and therefore less numbers in the older group?)  I think there needs to be a couple references here.  Perhaps the authors could consider approximating the prevalence of dextrocardia true prevalence as approximately 0.2/1,000?  That might clear up any confusion. 

- Line 532 – “Echocardiograms and Angiocardiograms” – should be changed to Echocardiograms and Angiography.

- Line 581 – “Angiocardiography” should be changed to angiography. 

Author Response

The reviewer comments “The authors have written a very clear, thorough and well-illustrated review of segmental analysis and a brief overview of management of various conditions associated with dextrocardia.  The senior author is particularly experienced and knowledgeable about segmental analysis, anomalies and dextrocardia conditions in particular.” - The author thanks the reviewer for the complimentary remarks.

The reviewer also states “The strengths of this manuscript are the organized, clear writing and the well described process of determining the cardiac segmental anatomy for a precise and accurate diagnosis.  I think that the writing style makes this appealing and read-able for cardiologists and non-cardiologists alike.  The illustrations are excellent and plentiful which add a great deal of value to the content.” - Thanks to the reviewer again.  

The reviewer further states “As a cardiologist, I think the weakness of this manuscript is simply the very broad nature of the review.  I would suggest that the editors consider limiting this review to only the diagnosis of dextrocardia and a pictorial rendition of segmental analysis and terminology.  The article is quite long.  The section on management is not very specific to dextrocardia since the variety of physiology spans the entire spectrum of single ventricle and biventricular disease.  The authors have done an admirable job summarizing different strategies, but the summary is essential the entire management of CHD.  I personally do not think the management section adds much and could be eliminated to keep the focus on the strong experience and knowledge of the authors on segmental analysis.”  - The reviewer suggests that the section on management may be deleted since the manuscript is long. It was deleted accordingly. Because of deleting the section on management, the title of the paper is changed to “Diagnosis of Dextrocardia With a Pictorial Rendition of Terminology and Diagnosis”

The reviewer summarizes “Overall, I think this is a very strong review of segmental analysis and I applaud the authors on their clear, detailed and thorough writing of an inherently complicated topic.” – Thanks. 

The reviewer makes “A few specific details and comments.” – I will revise/respond to each of these:  

- Figure 1 and 2 are redundant and one could be eliminated. – Figure 2 is deleted as recommended; the remaining figures are re-numbered.

- 3.9 Isolated Dextrocardia – seems redundant with the Dextrocardia section.  Is this necessary?  Consider eliminating this subsection. – I beg to disagree; Isolated Dextrocardia is an important condition.

- 3.10 Isolated Levocardia – also seems redundant and potentially confusing.  I would consider simply eliminating this sub-section. - I beg to disagree; Isolated Levocardia is also an important condition.

- 3.11 Other Terms – given the length of this manuscript, may be worth cutting this section which doesn’t seem to add anything essential.  – I have deleted this section, as recommended by this reviewer.

- Line 168 – This is confusing to me.  How can the prevalence of dextrocardia be lower in an older age group and higher in neonates?  Dextrocardia isn’t a condition that changes over time, obviously.  Are the numbers implying some attrition (i.e. - death of neonates with dextrocardia and therefore less numbers in the older group?)  I think there needs to be a couple references here.  Perhaps the authors could consider approximating the prevalence of dextrocardia true prevalence as approximately 0.2/1,000?  That might clear up any confusion. – The studies are from two different cohorts performed at different time periods. The explanation is likely that some dextrocardia neonates with complex heart disease may not have survived as the reviewer suggested. I do not believe a change in the script is necessary.

- Line 532 – “Echocardiograms and Angiocardiograms” – should be changed to Echocardiograms and Angiography. - Done

- Line 581 – “Angiocardiography” should be changed to angiography. – Done

The authors thank the reviewer for the diligent review and constructive criticism.

Reviewer 2 Report

This is a review article addressing the diagnosis and management of patients with dextrocardia including an overview and application of segmental analysis. Its strengths include that it is thoroughly written with abundant examples and figures. It provides a useful guideline for a trainee or general cardiologist in understanding segmental analysis and its application. Its weaknesses are a lack of clarity of the added value of this review, verbose and sometimes repetitive writing, the inclusion of poor-quality figures, and some outdated information.

The topic of segmental analysis has been thoroughly explained in publications by the same author as suggested in the list of references. The authors suggest that the added value of this review article is to provide an update on the approach to the diagnosis and management of cardiac malpositions. With regards to an update on the diagnosis, it is not clearly stated by the authors what information is new. Over 50% of the citations referenced were written by the senior author including works addressing cardiac malposition as recently as 2021 and 2022. It is difficult to discern what is new in this review that has not been recently published by the same author. Additionally, despite a stated desire to provide an updated review, there is a brief mention of newer imaging techniques (CT/MRI) but there are no figures dedicated to these. These imaging modalities have become a core component of the evaluation of children with cardiac malposition. With regard to updated information on management, the authors provide a very high-level overview of patient management with some outdated information. For example, the authors make a recommendation for prophylactic Ladd’s procedure and current data do not support routine upper GI screening or routine prophylactic Ladd procedure for the asymptomatic patient. If the authors believe otherwise, references should be provided.

All in all, the readers of this journal may benefit from a review of cardiac malposition, which is why I would recommend reconsidering after major revision which may include: clarifying what information is ‘updated’ in this review as stated by the authors, proofreading of grammatical errors (some mentioned below), more rigorous selection of figures (i.e. eliminate or replace poor-quality x-rays), and either eliminating the management section or providing a more robust review. Please find additional detailed comments below:

  • The first lines of the introduction could be modified to reflect the objective of the review as it pertains to and benefits the reader, rather than focusing solely on the previous work done by the author 

  • Line 38 - the use of the word “flabbergasted” seems too informal for use in an academic journal and would recommend changing it

  • Figure 2 seems superfluous. The chest x-ray is of poor quality (or at least appears that way on the document I was sent) and does not provide additional information to that portrayed in figure 1

  • Line 62 - the word “to” is missing after the word “secondary”

  • In figure 5 - is it necessary to use the word “roentgenogram” instead of the more commonly used term “x-ray”?

  • Figure 6 contains a poor quality x-ray

  • Line 103 - the word “is” is missing before the word “pushed”

  • Figure 15B - should read “left side of the chest” instead of the heart

  • The concept of chirality could be better explained through a figure. Readers who are not familiar with this method or have not thought it about it in some time, may not find it intuitive to think about right-hand and left-hand topology

  • In lines 810-811 - there is redundancy in stating both “avoiding hypothermia” and “preserving neutral thermal environment”

Author Response

The reviewer comments “This is a review article addressing the diagnosis and management of patients with dextrocardia including an overview and application of segmental analysis. Its strengths include that it is thoroughly written with abundant examples and figures. It provides a useful guideline for a trainee or general cardiologist in understanding segmental analysis and its application. Its weaknesses are a lack of clarity of the added value of this review, verbose and sometimes repetitive writing, the inclusion of poor-quality figures, and some outdated information.” – Thanks for the remarks. I will during the revision eliminate poor quality figures and ensure deleting repetitive sentences, as appropriate.

The reviewer also comments “The topic of segmental analysis has been thoroughly explained in publications by the same author as suggested in the list of references. The authors suggest that the added value of this review article is to provide an update on the approach to the diagnosis and management of cardiac malpositions. With regards to an update on the diagnosis, it is not clearly stated by the authors what information is new. Over 50% of the citations referenced were written by the senior author including works addressing cardiac malposition as recently as 2021 and 2022. It is difficult to discern what is new in this review that has not been recently published by the same author. Additionally, despite a stated desire to provide an updated review, there is a brief mention of newer imaging techniques (CT/MRI) but there are no figures dedicated to these. These imaging modalities have become a core component of the evaluation of children with cardiac malposition. With regard to updated information on management, the authors provide a very high-level overview of patient management with some outdated information. For example, the authors make a recommendation for prophylactic Ladd’s procedure and current data do not support routine upper GI screening or routine prophylactic Ladd procedure for the asymptomatic patient. If the authors believe otherwise, references should be provided.” – The reviewer takes exception to use of “updated review”. Accordingly, that is removed. Also the section on management was also removed.

The reviewer suggests “All in all, the readers of this journal may benefit from a review of cardiac malposition, which is why I would recommend reconsidering after major revision which may include: clarifying what information is ‘updated’ in this review as stated by the authors, proofreading of grammatical errors (some mentioned below), more rigorous selection of figures (i.e. eliminate or replace poor-quality x-rays), and either eliminating the management section or providing a more robust review.” – The section dealing with management is removed and other suggestions made by the reviewer were incorporated into the revised manuscript. Because of deleting the section on management, the title of the paper is changed to “Diagnosis of Dextrocardia With a Pictorial Rendition of Terminology and Diagnosis”

The reviewer makes additional suggestions: Please find additional detailed comments below: - these are addressed hereunder.

  • The first lines of the introduction could be modified to reflect the objective of the review as it pertains to and benefits the reader, rather than focusing solely on the previous work done by the author – as mentioned above, the objectives of the paper are revised, deleting “updated” as mentioned above.
  • Line 38 - the use of the word “flabbergasted” seems too informal for use in an academic journal and would recommend changing it – revised accordingly.
  • Figure 2 seems superfluous. The chest x-ray is of poor quality (or at least appears that way on the document I was sent) and does not provide additional information to that portrayed in figure 1 - Figure 2 is deleted as recommended; the remaining figures are re-numbered.
  • Line 62 - the word “to” is missing after the word “secondary” - Revised
  • In figure 5 - is it necessary to use the word “roentgenogram” instead of the more commonly used term “x-ray”? – Revised as recommended.
  • Figure 6 contains a poor-quality x-ray - - Figure 6 is deleted as recommended; the remaining figures are re-numbered.
  • Line 103 - the word “is” is missing before the word “pushed” - Revised as recommended.
  • Figure 15B - should read “left side of the chest” instead of the heart - Revised as recommended.
  • The concept of chirality could be better explained through a figure. Readers who are not familiar with this method or have not thought it about it in some time, may not find it intuitive to think about right-hand and left-hand topology – Agree.
  • In lines 810-811 - there is redundancy in stating both “avoiding hypothermia” and “preserving neutral thermal environment” – Agree; however, the section on management was deleted as suggested by both reviewers.

The authors thank the reviewer for the diligent review and constructive criticism.

Round 2

Reviewer 2 Report

The authors have made significant revisions to the manuscript. This will be a great resource for medical professionals involved in the care of children with congenital heart disease.